# ReciNet: Reciprocal Space-Aware Long-Range Modeling for Crystalline Property Prediction

**Jianan Nie**                                                          *jianan@vt.edu*
*Department of Computer Science*
*Virginia Tech*

**Peiyao Xiao**                                                         *peiyaoxi@buffalo.edu*
*Department of Computer Science and Engineering*
*University at Buffalo*

**Kaiyi Ji**                                                            *kaiyiji@buffalo.edu*
*Department of Computer Science and Engineering*
*University at Buffalo*

**Peng Gao**                                                            *penggao@vt.edu*
*Department of Computer Science*
*Virginia Tech*

**Reviewed on OpenReview:** *https://openreview.net/forum?id=ODlxgod5e3*

## Abstract

Predicting properties of crystals from their structures is a fundamental yet challenging task in materials science. Unlike molecules, crystal structures exhibit infinite periodic arrangements of atoms, requiring methods capable of capturing both local and global information effectively. However, current works fall short of capturing long-range interactions within periodic structures. To address this, we leverage *reciprocal space*, the natural domain for periodic crystals, and construct a Fourier series representation from fractional coordinates and reciprocal lattice vectors with learnable filters. Building on this, we introduce the reciprocal space-based geometry network (**ReciNet**), a novel architecture that integrates geometric GNNs and reciprocal blocks to model short-range and long-range interactions. Experiments on comprehensive benchmarks JARVIS, Materials Project, and MatBench demonstrate that ReciNet achieves outstanding predictive accuracy across a range of crystal property prediction tasks. Additionally, we explore a model extension for multi-property prediction with the mixture-of-experts, which demonstrates high computational efficiency and reveals positive transfer between correlated properties. These findings highlight the potential of our model as a scalable and accurate solution for crystal property prediction.

## 1 Introduction

AI-driven methods are revolutionizing materials science (Choudhary et al., 2022) by accelerating the discovery of novel materials that are critical for technological applications, including semiconductors, catalysts, and pharmaceuticals (Ashcroft & Mermin, 2001). In parallel, although traditional computational approaches, such as density functional theory (DFT), have provided significant advances, their high computational cost limits their scalability for exploring the vast materials space (Jones, 2015). To overcome these challenges, machine learning (ML) models, such as Graph Neural Networks (GNNs), have become prevailing for predicting material properties directly from their atomic structures (Chen et al., 2019; Louis et al., 2020; Zeni et al., 2023; Lin et al., 2023; Yan et al., 2024). However, these models suffer from a fundamental limitation

---

Code available at: https://github.com/peng-gao-lab/ReciNet

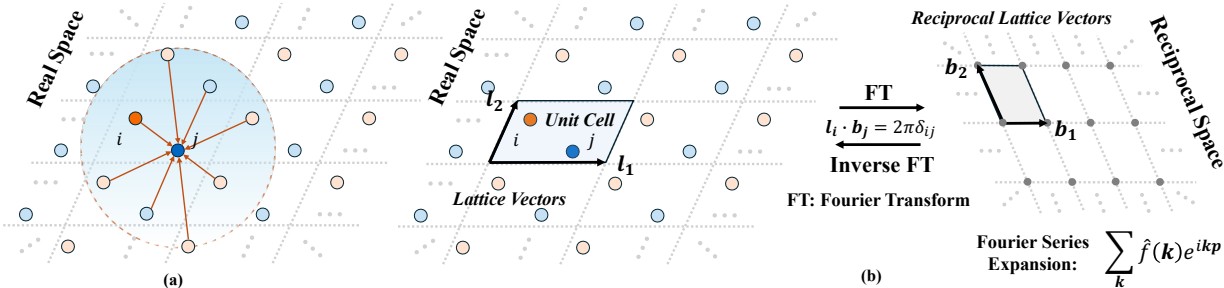

Figure 1: Schematic illustrations of capturing interactions in real space and reciprocal space. Note that we model 3D crystals, while we have a 2D illustration for simplicity. (a) An example crystal where each unit cell contains two atoms $i$ and $j$. Methods that rely solely on local information, such as CGCNN(Xie & Grossman, 2018), apply a fixed cutoff radius, thereby neglecting long-range interatomic interactions that are essential for accurately predicting crystal properties. (b) Relationship between real space (left) and reciprocal space (right). Reciprocal space enables efficient computation of long-range interactions via Fourier series expansion. See Section 3.2 for additional background.

that most approaches only capture local information, which introduces a *locality bias*. Message passing is typically restricted to atoms within a fixed cutoff radius, without capturing crystalline *infinite, periodic* atomic arrangement, and the associated long-range information, as illustrated in Figure 1 (Xie & Grossman, 2018; Schütt et al., 2018; Yan et al., 2022; Choudhary & DeCost, 2021). This limitation is particularly problematic because many key material properties, such as electronic band structures and mechanical moduli, are strongly dependent on this **long-range information**, creating a critical mismatch between conventional GNN paradigms and the underlying physics of crystalline solids.

The natural and principled framework for describing periodicity and long-range phenomena in solid-state physics is **reciprocal space** (Economou, 2010). By applying a Fourier transform, interactions that decay slowly in real space are represented compactly in reciprocal space, where only a few strong, low-frequency components dominate. This transformation makes modeling long-range interactions computationally tractable and provides a physically grounded strategy for overcoming the locality bias. *However, designing an effective, end-to-end machine learning architecture that leverages reciprocal space for crystalline systems has remained a significant and largely unmet challenge.*

A few early attempts have been made to integrate reciprocal space for crystalline property prediction, but these preliminary efforts fall short due to critical design limitations. One line of work models long-range interactions through *fixed, non-trainable* formulations. PotNet (Lin et al., 2023) encodes precomputed physical potentials derived from Ewald summation, while Crystalformer (Taniai et al., 2024) incorporates a dual-space attention mechanism based on analytical Fourier-Gaussian kernels. However, these methods rely on rigid, handcrafted functions that prevent the model from learning material-specific reciprocal space interactions. Consequently, they provide only a shallow integration of reciprocal space and fail to fully leverage the symmetry-aware structure intrinsic to periodic crystals. A separate line of work, while offering a learnable design, suffers from other fundamental shortcomings. EwaldMP (Kosmala et al., 2023) introduces an end-to-end approach with structure factor embeddings. However, it was developed primarily for molecular systems and relies on an artificial supercell grid. This formulation is fundamentally *misaligned with the physics of crystalline solids*, as it breaks the crystal's inherent space group symmetries and introduces computationally costly grid-size hyperparameters. Together, these issues highlight a clear and critical gap: *the lack of a crystal-native framework that simultaneously preserves intrinsic periodic symmetries and adaptively learns complex interactions in reciprocal space.*

In this work, we bridge this gap by introducing the *reciprocal space-based geometry network (**ReciNet**)*, a neural network architecture designed to capture long-range interactions in a crystal-native and system-adaptive manner. Our key contribution lies in the reciprocal block, a novel module that models long-range interactions by constructing a learnable Fourier series representation from fractional coordinates and

reciprocal lattice vectors. This design inherently preserves crystal symmetries and adapts to diverse material systems without relying on fixed grids or analytical functions. Then, ReciNet integrates this reciprocal block with a local geometric GNN, forming a hybrid architecture that jointly models short-range chemical bonding and long-range lattice-scale effects. We further extend the framework to a multi-task variant, enabling more efficient and scalable multi-property prediction. Finally, extensive experiments show that ReciNet achieves superior performance compared to various methods across common material property prediction datasets, JARVIS (Choudhary et al., 2020), the Materials Project (Chen et al., 2019), and MatBench datasets (Dunn et al., 2020). Especially, ReciNet achieves 13.7% lower MAE in bulk modulus prediction and 7.7% in band gap (MBJ) prediction compared with iComFormer. Lastly, we show the superiority of our method compared to prior methods that incorporate reciprocal space information.

## 2 Related Work

**Crystal property prediction.** Crystal property prediction has been widely studied using both physics-based and deep-learning approaches. Traditional physics-based methods, such as Coulomb matrices (Rupp et al., 2012; Elton et al., 2018), effectively model ionic and metallic materials but lack generalization due to constraints like permutation invariance. Deep learning has introduced more flexible predictions by representing crystals as chemical formulas and using sequence models (Jha et al., 2018; Goodall & Lee, 2020; Wang et al., 2021). More recent methods leverage 3D geometric structures, modeling as 3D graphs in SchNet (Schütt et al., 2017) and DimeNet (Gasteiger et al., 2020). Key advances include CGCNN (Xie & Grossman, 2018), which utilizes multi-edge graphs to model periodic invariance. ALIGNN (Choudhary & DeCost, 2021) incorporates angle-based features, and Matformer (Yan et al., 2022) captures periodic patterns using self-connecting edges. Yan et al. (2024) introduces iComFormer with invariant descriptors and eComFormer with equivariant vectors. CrystalFramer (Ito et al., 2025) models SE(3)-invariant features using dynamic, atomwise coordinate frames. Despite advancements, these approaches face significant limitations in accurately capturing periodic patterns and long-range interactions.

**Long-range interaction modeling.** Modeling long-range interactions in periodic crystals remains a significant challenge. Early data-driven methods augmented local message-passing networks with simplified global information, such as predicting atomic charges (Unke et al., 2021) or integrating computed scalar-valued structure factors only at the final layer (Yu et al., 2022), which limits the rich interplay between local and global features. More physically-grounded approaches operate in reciprocal space, the natural domain for periodic systems (Ashcroft & Mermin, 2001). However, these models often suffer from the following critical limitations. PotNet (Lin et al., 2023) uses pre-computed potential fields derived from Ewald summation as static node features, while Crystalformer (Taniai et al., 2024) employs an attention mechanism based on fixed, analytical Fourier-Gaussian kernels. While physically motivated, these hand-crafted functions are universal across all materials. Moreover, the pre-computed potential in PotNet requires high $\mathcal{O}(n^2)$ computational cost, where $n$ is the number of atoms in the unit cell. Another approach adopts a non-crystal-native formulation. EwaldMP (Kosmala et al., 2023), for example, computes learnable structure factor embeddings but then projects them onto an artificial supercell grid to perform reciprocal-space message passing. This grid-based approach is fundamentally misaligned with true crystal periodicity, as the arbitrary choice of grid size can break the crystal's intrinsic space group symmetries and introduce computationally expensive hyperparameters. We include a direct comparison with reciprocal space methods in Section 5.1.

## 3 Preliminary

### 3.1 Periodic Crystal Structure Representation

A crystalline material can be represented as the infinite periodic arrangement of atoms in 3D space, with the smallest repeating structural unit referred to as the unit cell (Ashcroft & Mermin, 2001). As the fundamental repeating unit, the unit cell contains the atomic arrangement and lattice vectors that establish a crystal's periodicity and space group symmetry, which together dictate all of its material properties. The structure of crystals is commonly represented using two coordinate systems: the Cartesian coordinate system and the fractional coordinate system.

**Cartesian coordinate.** Mathematically, a material can be represented as $\mathbf{M} = (\mathbf{A}, \mathbf{P}, \mathbf{L})$. $\mathbf{A} = [\boldsymbol{a}_1, \boldsymbol{a}_2, \cdots, \boldsymbol{a}_n]^\top \in \mathbb{R}^{n \times h}$ denotes the atom feature matrix for $n$ atoms within a unit cell, where $\boldsymbol{a}_i \in \mathbb{R}^h$ represents the $h$-dimensional features of atom $i$ in the unit cell. $\mathbf{P} = [\boldsymbol{p}_1, \boldsymbol{p}_2, \cdots, \boldsymbol{p}_n]^\top \in \mathbb{R}^{n \times 3}$ is the position matrix, where $\boldsymbol{p}_i \in \mathbb{R}^3$ is the Cartesian coordinate of atom $i$ inside the unit cell. In crystallography, a lattice is a periodic arrangement of points that defines a crystal's translational symmetry. As shown in Figure 1, the lattice matrix $\mathbf{L} = [\boldsymbol{\ell}_1, \boldsymbol{\ell}_2, \boldsymbol{\ell}_3]^\top \in \mathbb{R}^{3 \times 3}$ consists of three translation lattice vectors that determine the shape and periodicity of the unit cell. The periodic repetition of the unit cell is described by integer multiples of the lattice vectors, which establishes the crystal's inherent translational symmetry. Specifically, the infinite crystal structure can be expressed:

$$\hat{\mathbf{P}} = \{\hat{\boldsymbol{p}}_i | \hat{\boldsymbol{p}}_i = \boldsymbol{p}_i + n_1\boldsymbol{\ell}_1 + n_2\boldsymbol{\ell}_2 + n_3\boldsymbol{\ell}_3, n_1, n_2, n_3 \in \mathbb{Z}, i \in \mathbb{Z}, 1 \leq i \leq n\} \tag{1}$$

where the integers $n_1, n_2, n_3$ define a 3D translation with $\boldsymbol{\ell}_1, \boldsymbol{\ell}_2, \boldsymbol{\ell}_3$. $\hat{\mathbf{A}} = \{\hat{\boldsymbol{a}}_i | \hat{\boldsymbol{a}}_i = \boldsymbol{a}_i, i \in \mathbb{Z}, 1 \leq i \leq n\}$ is the atom feature in repeated unit cells, which remains unchanged under periodic translation. In our framework, Cartesian coordinates are used for real space graph construction, where accurate interatomic distances are needed for radius-based cutoff and edge feature initialization.

**Fractional coordinate.** Fractional coordinates capture the relative positions of atoms within the unit cell and offer significant advantages for periodic materials compared to Cartesian coordinates. Specifically, the fractional coordinate uses the lattice matrix $\mathbf{L} \in \mathbb{R}^{3 \times 3}$ as the basis, where atomic positions are described by fractional coordinate vectors $\boldsymbol{f}_i = [f_1, f_2, f_3]^\top \in [0, 1)^3$. The corresponding Cartesian coordinates are obtained as $\boldsymbol{p}_i = \boldsymbol{L}\boldsymbol{f}_i$, where $\boldsymbol{p}_i = f_1\boldsymbol{\ell}_1 + f_2\boldsymbol{\ell}_2 + f_3\boldsymbol{\ell}_3$. The representation of crystal $\mathbf{M}$ generalizes to $\mathbf{M} = (\mathbf{A}, \mathbf{F}, \mathbf{L})$, where $\mathbf{F} = [\boldsymbol{f}_1, \cdots, \boldsymbol{f}_n]^\top \in [0, 1)^{n \times 3}$ denotes the fractional coordinates of all atoms in the unit cell. Fractional coordinates inherently align with lattice periodicity, ensuring that calculations respect periodic boundary conditions. Also, they maintain structural relationships under space group operations, such as translations and rotations, consistently preserving the crystal's underlying symmetry. These properties establish fractional coordinates as a more efficient and symmetry-aware representation for periodic materials compared to Cartesian coordinates, making them essential for the reciprocal space modeling.

### 3.2 Reciprocal Space in Crystalline Materials

The periodic arrangement of atoms in crystalline materials fundamentally influences their physical properties by establishing a repeating structure that extends over long distances. *Reciprocal space* provides a mathematical framework to describe the periodic structure of crystals, enabling the analysis and prediction of material properties influenced by periodicity (Ashcroft & Mermin, 2001; Gross & Marx, 2014; Economou, 2010). Reciprocal space consists of wave vectors $\boldsymbol{k}$ spanned by reciprocal lattice vectors $\boldsymbol{b_1}, \boldsymbol{b_2}, \boldsymbol{b_3}$, which are derived from lattice vectors in real space $\boldsymbol{\ell}_1, \boldsymbol{\ell}_2, \boldsymbol{\ell}_3$, with relationship satisfying $\boldsymbol{b_i} \cdot \boldsymbol{\ell_j} = 2\pi\delta_{ij}$, where $\delta_{ij}$ is the Kronecker delta. Each vector in reciprocal space represents a spatial frequency that encodes crystal periodicity along specific directions. More details can be found in Appendix A.

To establish the reciprocal space representation, we can use the Fourier transform, which expresses the periodic function $f$ via Fourier series expansion:

$$f(\boldsymbol{p}_j) = \sum_{\boldsymbol{k}} \hat{f}(\boldsymbol{k}) \exp(i\boldsymbol{k} \cdot \boldsymbol{p}_j), \tag{2}$$

where $\hat{f}(\boldsymbol{k})$ represents the Fourier coefficients, encoding the contributions of wave vectors $\boldsymbol{k}$ at atom $j$. The corresponding inverse transform, which computes $\hat{f}(\boldsymbol{k})$ from the real space representation $f(\boldsymbol{p}_j)$, is given by:

$$\hat{f}(\boldsymbol{k}) = \frac{1}{\Omega} \sum_{j \in S} h_j \exp(-i\boldsymbol{k} \cdot \boldsymbol{p}_j), \tag{3}$$

where $S$ denotes the set of atomic positions in the crystal lattice, $h_j$ represents function values at site $\boldsymbol{p}_j$ such as nuclear embedding, and $\Omega$ is the system volume.

# 4 Method

Although recent methods have made progress in modeling long-range interactions for crystal property prediction, they often rely on fixed physical formulations (Lin et al., 2023; Taniai et al., 2024) or predefined supercell grids (Kosmala et al., 2023), which limit their adaptability and may break crystal symmetries. In this work, we propose ReciNet, a novel architecture that overcomes these limitations by introducing a *learnable* reciprocal space block to model long-range periodic interactions. Our proposed ReciprocalBlock, which operates *directly on fractional coordinates and reciprocal lattice vectors*, learns continuous Fourier-space filters that preserve periodicity and space group symmetries without requiring supercell construction or fixed analytic functions. It operates as a *hybrid architecture*, synergistically integrating a geometric GNN for short-range effects with the ReciprocalBlock for long-range interactions. For more efficient property prediction, we introduce ReciNet-MT, a multi-task variant that incorporates a task-aware mixture-of-experts (MoE) decoder. The overall architectures of ReciNet and ReciNet-MT are illustrated in Figure 2.

## 4.1 Representation and Embedding

**Crystal graph and embedding.** Firstly, we construct a radius-based graph, where nodes represent atoms and edges encode atomic interactions within a predefined cutoff radius $r_{\text{cut}}$. Formally, the edge set is defined as:

$$\mathcal{E} = \{e_{ij} : \|\boldsymbol{p}_i - \boldsymbol{p}_j\|_2 \leq r_{\text{cut}}, \ \forall i, j \in V\}, \qquad (4)$$

where $V$ is the set of nodes, and distances between nodes are computed based on their 3D Cartesian coordinates $\boldsymbol{p}_i$ and $\boldsymbol{p}_j$.

Once the crystal graph is established, we can proceed to obtain embeddings for nodes and edges. Node feature $\boldsymbol{a}_i$ for node $i$ is first mapped using CGCNN (Xie & Grossman, 2018) embeddings, followed by a linear transformation to initialize the short-range node features $h^0_{i,\text{local}}$. The initial global atom representation is derived through a linear transformation of $h^0_{\text{local}}$, as shown below:

$$h^0_{\text{global}} = \sigma(W_r h^0_{\text{local}}), \qquad (5)$$

where $W_r$ is a learnable weight matrix. We use $h^0_{\text{global}}$ as initial long-range node features. The edge

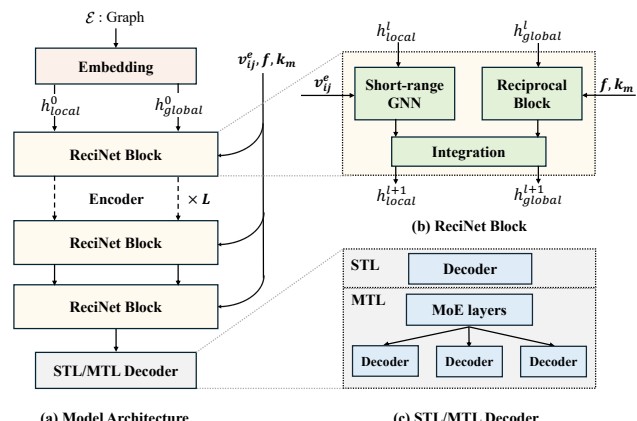

Figure 2: Overall framework architecture, consisting of three components: (a) Model architecture, (b) ReciNet block, (c) Decoder. The model uses stacked blocks, which integrate local features via message passing and global features via reciprocal space convolution. The decoder outputs property predictions for single-task and multi-task settings.

features $\|\mathbf{p}_i - \mathbf{p}_j\|_2$, defined by the Euclidean distance between node pair $i$ and $j$, are scaled by $c/\|\mathbf{p}_i - \mathbf{p}_j\|_2$, where c is a chosen constant to mimic the pairwise potential in Lin et al. (2023). These values are then embedded using radial basis function (RBF) kernels to get the initial edge features $v^e_{ij}$. More details are shown in Appendix B.2

**Fractional coordinates and reciprocal lattice vectors.** To effectively extract features from long-range information in reciprocal space, we propose using fractional coordinates $\boldsymbol{f}$ and reciprocal lattice vectors $\boldsymbol{k}$ (Section 3) with continuous filters convolution. This approach differs from previous methods for the following reasons. First, for crystalline materials, Cartesian coordinates can introduce ambiguities, as identical crystal structures may be expressed differently due to periodic transformations, such as translations or rotations. For example, the lattice matrices $\mathbf{L} = [\boldsymbol{\ell}_1, \boldsymbol{\ell}_2, \boldsymbol{\ell}_3]^\top$ and $\mathbf{L}' = [\boldsymbol{\ell}_1 + \boldsymbol{\ell}_2, \boldsymbol{\ell}_2, \boldsymbol{\ell}_3]^\top$ describe identical periodic patterns but differ in their Cartesian representations. In contrast, fractional coordinates normalize atomic positions relative to the unit cell, ensuring consistent representation under periodic transformations and eliminating redundancies in Cartesian representations. Second, Kosmala et al. (2023) relies on a predefined grid size as hyperparameter to cover all relevant frequencies, which could disrupt the unit cell symmetry and

Table 1: **Comparison of test MAE on the Materials Project dataset.** The best results are highlighted in **bold**, while the second-best results are indicated with underlines.

| Method | Formation Energy | Band Gap | Bulk Moduli | Shear Moduli |
|---|---|---|---|---|
| | meV/atom | eV | log(GPa) | log(GPa) |
| CGCNN (Xie & Grossman, 2018) | 31 | 0.292 | 0.047 | 0.077 |
| SchNet (Schütt et al., 2017) | 33 | 0.345 | 0.066 | 0.099 |
| MEGNET (Chen et al., 2019) | 30 | 0.307 | 0.060 | 0.099 |
| GATGNN (Louis et al., 2020) | 33 | 0.280 | 0.045 | 0.075 |
| ALIGNN (Choudhary & DeCost, 2021) | 22 | 0.218 | 0.051 | 0.078 |
| Matformer (Yan et al., 2022) | 21.0 | 0.211 | 0.043 | 0.073 |
| PotNet (Lin et al., 2023) | 18.8 | 0.204 | 0.040 | 0.0650 |
| Crystalformer (Taniai et al., 2024) | 18.6 | 0.198 | 0.0377 | 0.0689 |
| eComFormer (Yan et al., 2024) | 18.16 | 0.202 | 0.0417 | 0.0729 |
| iComFormer (Yan et al., 2024) | 18.26 | 0.193 | 0.0380 | 0.0637 |
| CrystalFramer (Ito et al., 2025) | 17.2 | **0.185** | 0.0338 | 0.0677 |
| **ReciNet** | **17.07** | 0.189 | **0.0328** | **0.0628** |

substantially increase computational costs, particularly for large unit cells and complex systems. Instead, by combining fractional coordinates $\boldsymbol{f}$ and basis reciprocal lattice vectors $\boldsymbol{k}_m$, we achieve an effective framework for feature extraction in reciprocal space, accurately capturing the long-range information of crystals.

With the aforementioned representations of crystalline materials, our model enables both short-range and long-range message passing to effectively capture comprehensive structural features.

## 4.2 Short-Range Message Passing

We employ the short-range block in Figure 2 to update the atomic representations using the graph neural network with multiple message-passing layers. Each node aggregates information only from its geometric neighbors within the cutoff radius $r_{\text{cut}}$. Therefore, we naturally interpret this as capturing the short-range information in material representations. The computational process at the $\ell$-th message-passing layer for a node $i$ is expressed as:

$$h_{i,\text{local}}^{\ell+1} \leftarrow g\left(h_{i,\text{local}}^{\ell}, \sum_{j \in V} \phi\left(h_{i,\text{local}}^{\ell}, h_{j,\text{local}}^{\ell}, v_{ij}^{e}\right)\right), \tag{6}$$

where $h_{i,\text{local}}^{\ell}$ represents the embedding features of node $i$ at the $\ell$-th layer. Here, $g$ and $\phi$ are trainable layers within the GNN. Specifically, $\phi$ computes interactions between node embeddings and edge features $v_{ij}^{e}$, while $g$ aggregates these messages to update the node embeddings. Further details on the architecture of the local geometric GNN are provided in Appendix B.

## 4.3 Long-Range Message Passing

To complement the short-range modeling, we employ the long-range block in Figure 2 to globally update the atomic representations using the *ReciprocalBlock*. Specifically, this block leverages Fourier transform-based continuous filters with neural networks to encode periodic structures from reciprocal space. Besides, as discussed in Section 4.1, we utilize atomic fractional coordinates $\boldsymbol{f}$ and basis reciprocal lattice vectors $\boldsymbol{k}_m$ within material $m$ to extract structural information from reciprocal space and update the node features in the model block. The global embeddings are computed iteratively:

$$h_{\text{global}}^{\ell+1} = \text{ReciprocalBlock}(h_{\text{global}}^{\ell}, \boldsymbol{f}, \boldsymbol{k}_m), \tag{7}$$

where $h_{\text{global}}^{\ell}$ represents the input embeddings from the $\ell$-th layer.

**Reciprocal block.** Here, we explain the reciprocal block in detail. The reciprocal block incorporates long-range interactions into the node embeddings using continuous filters on features derived from Fourier

transforms. Each material $m$ has reciprocal lattice vectors $\boldsymbol{k}_m$ based on the lattice vector in real space. The set of atoms in the material is denoted as $\mathcal{I}_m$, where $j \in \mathcal{I}_m$ represents all atoms within material $m$. This block aggregates the contributions of all atoms in each material to compute a global representation in the reciprocal domain. The reciprocal embedding $r_m$ is computed as:

$$r_m = \sum_{j \in \mathcal{I}_m} h_{j,\text{global}}^{\ell} \cdot \exp(-i\boldsymbol{k}_m^{\top} \boldsymbol{f}_j), \tag{8}$$

where $h_{j,\text{global}}^{\ell}$ is the global embedding of atom $j$ from $\ell$-th layer, $\boldsymbol{f}_j$ is its fractional coordinate, and $i$ is the imaginary unit. This operation aggregates atomic embeddings into a single reciprocal representation for materials, capturing global periodic interactions.

Subsequently, the reciprocal embedding $r_m$ undergoes an inverse Fourier transform to be mapped back into real space for the atoms in material $m$. This process can be expressed as follows:

$$\tilde{h}_{\text{global}}^{\ell} = \sum_{j \in \mathcal{I}_m} \exp(i\boldsymbol{k}_m^{\top} \boldsymbol{f}_j) \cdot r_m \cdot \mathbf{W}_{\text{filter}}, \tag{9}$$

where $\exp(i\boldsymbol{k}_m^{\top} \boldsymbol{f}_j)$ performs the inverse Fourier transform to map the reciprocal representation back to the real domain. $\mathbf{W}_{\text{filter}}$ is a *trainable filter* refining the reciprocal embedding, selectively emphasizing important contributions from the reciprocal space features to model long-range interactions. The prefactor $\frac{1}{\Omega}$ in eq. (3) is incorporated into the learned filter $\mathbf{W}_{\text{filter}}$, without requiring explicit normalization by the system volume. In a word, this operation effectively incorporates long-range interactions from the reciprocal domain into the real space representation of each atom $j \in \mathcal{I}_m$.

Finally, the global embeddings $\tilde{h}_{\text{global}}^{\ell}$ are updated using residual connections to get $h_{\text{global}}^{\ell+1}$.

$$h_{\text{global}}^{\ell+1} = h_{\text{global}}^{\ell} + \tilde{h}_{\text{global}}^{\ell}. \tag{10}$$

At this point, the complete process of a reciprocal block has been fully described.

### 4.4 Hierarchical Embedding Integration

Our model employs multiple ReciNet blocks as message passing layers to integrate local and global information at each layer. Within each block, node embeddings are updated by combining contributions from short-range graph message passing and long-range reciprocal-based message passing. The resulting integrated embeddings, formed by adding up both embedding updates during each message-passing step, are propagated to the next block. This enables the model to iteratively refine hierarchical representations of material structures. This design facilitates the effective capture of both local atomic-scale features and global lattice-scale features throughout the network.

### 4.5 Decoder Block

After completing the iterative message-passing steps, node features are aggregated within each graph using mean pooling, which captures the overall structural information of the material. The resulting graph-level representation is then processed through fully connected layers to predict the material property. While our novel reciprocal space representation effectively captures long-range interactions and global periodicity, it still predicts properties individually (Schütt et al., 2017; Choudhary & DeCost, 2021; Yan et al., 2024). To validate the representation ability of the ReciNet encoder, we also explore its application in a multi-task learning (MTL) setting. This variant, ReciNet-MT, couples the shared encoder with an efficient mixture-of-experts (MoE) decoder to predict multiple properties simultaneously, as shown in Figure 2. More details can be found in Appendix C.

## 5 Experimental Studies

We evaluate ReciNet on three widely recognized datasets in materials science, Materials Project (Chen et al., 2019), JARVIS-DFT (Choudhary et al., 2020), and MatBench (Dunn et al., 2020). Through extensive

Table 2: **Comparison of test MAE on the JARVIS dataset.** The best results are highlighted in **bold**, and the second-best results are indicated with underlines. Lower test MAE indicates better results.

| Method | Form. Energy | Bandgap(OPT) | $E_{\text{total}}$ | Bandgap(MBJ) | $E_{\text{hull}}$ |
|---|---|---|---|---|---|
| | meV/atom | eV | meV/atom | eV | meV |
| CGCNN (Xie & Grossman, 2018) | 63 | 0.20 | 78 | 0.41 | 170 |
| SchNet (Schütt et al., 2017) | 45 | 0.19 | 47 | 0.43 | 140 |
| MEGNET (Chen et al., 2019) | 47 | 0.145 | 58 | 0.34 | 84 |
| GATGNN (Louis et al., 2020) | 47 | 0.170 | 56 | 0.51 | 120 |
| ALIGNN (Choudhary & DeCost, 2021) | 33.1 | 0.142 | 37 | 0.31 | 76 |
| Matformer (Yan et al., 2022) | 32.5 | 0.137 | 35 | 0.30 | 64 |
| PotNet (Lin et al., 2023) | 29.4 | 0.127 | 32 | 0.27 | 55 |
| Crystalformer (Taniai et al., 2024) | 30.6 | 0.128 | 32 | 0.27 | 46 |
| eComFormer (Yan et al., 2024) | 28.4 | 0.124 | 32 | 0.28 | 44 |
| iComFormer (Yan et al., 2024) | 27.2 | **0.122** | 28.8 | 0.26 | 47 |
| **ReciNet** | **27.0** | 0.126 | **26.8** | **0.24** | **43** |

Table 3: **Comparison on MatBench.** The best results are highlighted in **bold**, and the second-best results are indicated with underlines. Lower test MAE and test RMSE indicate better results.

| Method | e_form-132752 (meV/atom) | | jdft2d-636 (meV/atom) | |
|---|---|---|---|---|
| | MAE | RMSE | MAE | RMSE |
| MODNet (De Breuck et al., 2021) | $44.8 \pm 3.9$ | $88.8 \pm 7.5$ | $33.2 \pm 7.3$ | $96.7 \pm 40.4$ |
| ALIGNN (Choudhary & DeCost, 2021) | $21.5 \pm 0.5$ | $55.4 \pm 5.5$ | $43.4 \pm 8.9$ | $117.4 \pm 42.9$ |
| coGN (Ruff et al., 2024) | $17.0 \pm 0.3$ | $48.3 \pm 5.9$ | $37.2 \pm 13.7$ | $101.2 \pm 55.0$ |
| M3GNet (Chen & Ong, 2022) | $19.5 \pm 0.2$ | - | $50.1 \pm 11.9$ | - |
| eComFormer (Yan et al., 2024) | $16.5 \pm 0.3$ | $45.4 \pm 4.7$ | $37.8 \pm 9.0$ | $102.2 \pm 46.4$ |
| iComFormer (Yan et al., 2024) | $16.5 \pm 0.3$ | $43.2 \pm 3.7$ | $34.8 \pm 9.9$ | $96.1 \pm 46.3$ |
| **ReciNet** | **$16.1 \pm 0.2$** | **$42.9 \pm 3.5$** | **$31.6 \pm 6.3$** | **$75.6 \pm 34.7$** |

experiments, ReciNet demonstrates strong performance across various crystal properties, highlighting its effectiveness in modeling crystalline materials with reciprocal space.

For JARVIS and MP, to ensure consistency, we adopt experimental settings aligned with prior works, including Matformer (Yan et al., 2022) and PotNet (Lin et al., 2023). These datasets cover a range of scales, including 69,239 crystals for large-scale tasks, 18,171 crystals for medium-scale tasks, and 5,450 crystals for small-scale tasks. To further assess performance, we evaluate ReciNet on the MatBench tasks `e_form` (132,752 crystals) and `jdft2d` (636 crystals), following the experimental setup in Yan et al. (2024). We benchmark against major baseline methods, including CGCNN (Xie & Grossman, 2018), SchNet (Schütt et al., 2017), MEGNet (Chen et al., 2019), GATGNN (Louis et al., 2020), ALIGNN (Choudhary & DeCost, 2021), Matformer (Yan et al., 2022), PotNet (Lin et al., 2023), Crystalformer (Taniai et al., 2024), ComFormer (Yan et al., 2024), CrystalFramer (Ito et al., 2025), MODNet (De Breuck et al., 2021) , coGN (Ruff et al., 2024), and M3GNet (Chen & Ong, 2022). We exclude CrystalFormer from the JARVIS comparison to ensure fairness, as it requires significantly longer training time than other baselines, as shown in Table 5. We also compare ReciNet with reciprocal space related methods, PotNet (Lin et al., 2023), EwaldMP (Kosmala et al., 2023), and Crystalformer (Taniai et al., 2024) variant, to highlight the advantage of ReciNet. Experiments are conducted on NVIDIA A100 GPUs. Additional details on the experimental configurations and implementation settings are provided in Appendix B.

## 5.1 Experimental Results

**The Materials Project (MP).** On the Materials Project dataset, ReciNet consistently outperforms prior methods across all four benchmark tasks, achieves the lowest MAE on formation energy, bulk modulus, and

Table 4: **Ablation studies.** Reciprocal space and model depth in test MAE on the JARVIS dataset.

| Method | Reciprocal | Blocks | Form. Energy | Bandgap(OPT) | $E_{total}$ | Bandgap(MBJ) | $E_{hull}$ |
|---|---|---|---|---|---|---|---|
| ReciNet w/o Reciprocal | ✗ | 3 | 29.6 | 0.135 | 32.5 | 0.295 | 68.8 |
| ReciNet w/o Reciprocal | ✗ | 4 | 28.9 | 0.132 | 31.0 | 0.290 | 65.6 |
| ReciNet | ✓ | 3 | 27.8 | 0.127 | 27.2 | 0.242 | 52.9 |
| ReciNet | ✓ | 4 | **27.0** | **0.126** | **26.8** | **0.239** | **48.0** |

shear modulus, and is highly competitive on band gap. As shown in Table 1, ReciNet achieves lower MAE than iComFormer on all target properties, including a 6.5% reduction in formation energy error, 2.1% in bandgap, and 13.7% in bulk modulus. Measures of variation are reported in Table 18.

**JARVIS-DFT.** As shown in Table 2, ReciNet surpasses baseline models in formation energy, $E_{\text{total}}$, and band gap (MBJ), while achieving competitive results on $E_{\text{hull}}$ and band gap (OPT). Notably, ReciNet consistently surpasses PotNet across all tasks and improves upon iComFormer with a 7.7% lower MAE on band gap (MBJ) and a 6.9% reduction on total energy. Measures of variation are reported in Table 17.

**MatBench.** Experimental results on the MatBench benchmark are summarized in Table 3, reporting MAE and RMSE with standard deviations over five runs. ReciNet achieves SOTA performance on both the large-scale `e_form` task (over 130K crystals) and the small-scale `jdft2d` task (636 crystals), demonstrating strong generalization across both large and limited training data. It outperforms models that incorporate geometric features such as bond angles, including ALIGNN and M3GNet, as well as coGN, which employs a connectivity-optimized nested graph representation. ReciNet also surpasses MODNet, a model based on physical descriptors with feature selection. These results highlight the effectiveness of incorporating reciprocal space information into crystal representations.

**Reciprocal space comparison.** To further demonstrate the superiority of reciprocal space design in ReciNet, we provide detailed comparisons with machine learning models that incorporate reciprocal space information, including PotNet, EwaldMP, and Crystalformer, as shown in Table 13. Detailed results and discussions are presented in Appendix B.6. In summary, ReciNet consistently achieves lower MAE across all five crystal properties on the JARVIS dataset compared to these baselines.

Overall, ReciNet achieves the best performance on 11 out of 13 crystal properties across three widely used benchmarks, while remaining within 3.3% of the best method on the remaining two. This success is rooted in an architectural design that captures the full hierarchy of physical interactions in crystals. ReciNet overcomes the locality bias by operating in reciprocal space, the natural domain for periodic systems. Specifically, the usage of fractional coordinates $f$ and reciprocal lattice vectors $k$ constructs a Fourier series representation of the crystal. This process adaptively learns long-range representation while reserving crystal-native symmetry.

## 5.2 Ablation Studies

We conduct ablation studies on the JARVIS dataset to examine the contribution of the reciprocal space.

**Effect of the reciprocal block.** We evaluate the ReciNet with the reciprocal space component removed. As shown in Table 4, including reciprocal space information leads to consistent improvement across five properties. Furthermore, adding one more reciprocal block yields additional gains, indicating that deeper reciprocal modeling captures richer long-range structure. Additional depth analysis is provided in Appendix B.5.

**Generalizability of the reciprocal block.** To verify that these gains are not specific to the ReciNet backbone, we further evaluate the ReciprocalBlock as a plug-and-play module by integrating it into CGCNN and Matformer. As shown in Appendix B.8, this addition yields consistent improvements across all five JARVIS properties for both backbones (e.g., Matformer formation energy MAE drops from 32.5 to 28.9 meV/atom with 3 reciprocal layers), confirming that the module is generally effective and not architecture-specific.

Table 5: **Efficiency comparison.** Comparison of model efficiency on JARVIS formation energy prediction. We include training time per epoch, training epochs, total training time, average inference time per material, and the number of parameters. N/A denotes that ReciNet-MT is not designed for single-property prediction.

| Model | Type | Time/Epoch | Epochs | Total | Inference | 1-task Para. | 5-task Para. |
|---|---|---|---|---|---|---|---|
| PotNet (Lin et al., 2023) | GNN | 43 s | 500 | 6.0 h | 313 ms | 1.8 M | 9.0 M |
| Matformer (Yan et al., 2022) | Transformer | 60 s | 500 | 8.3 h | 20.4 ms | 2.9 M | 14.5 M |
| iComFormer (Yan et al., 2024) | Transformer | 59 s | 700 | 11.5 h | 54.8 ms | 5.0 M | 25.0 M |
| Crystalformer (Taniai et al., 2024) | Transformer | 32 s | 800 | 7.2 h | 6.6 ms | 853 K | 4.27 M |
| CrystalFramer (Ito et al., 2025) | Transformer | 74 s | 2000 | 41.2 h | 16.8 ms | 952 K | 4.76 M |
| **ReciNet** | GNN | 35 s | 500 | **4.8 h** | 8.2 ms | 3.3 M | 16.5 M |
| **ReciNet-MT** | GNN | 69 s | 500 | 9.6 h | 16.2ms | N/A | 9.5 M |

## 5.3 Multi-property prediction

We include ReciNet-MT as an extension to demonstrate the scalability of our design in multi-property settings, with detailed descriptions and experiments in Appendix C. On the Materials Project dataset, we apply two ReciNet-MT models separately for predicting two energy properties and two mechanical properties. It achieves comparable performance as shown in Table 19 in the Appendix. For the JARVIS dataset, ReciNet-MT on 5-property prediction delivers the best results for band gap (OPT) and band gap (MBJ) while maintaining comparable performance on the remaining tasks to Matformer as shown in Table 14. The observed improvement can be attributed to positive transfer as discussed in Appendix C.5.

## 5.4 Model Efficiency Comparison

Table 5 reports the efficiency of top-performing architectures in terms of runtime and model size. ReciNet attains superior accuracy while requiring only 12% of the training time of CrystalFramer, 42% of iCom-Former, and 66% of Crystalformer. Although Crystalformer and CrystalFramer incur low per-epoch costs, they require a large number of training epochs (800 and 2,000, respectively) to converge. Given such a significant gap in computational cost, a direct performance comparison with other methods in the comparison would not be entirely fair. Thus, we exclude the CrystalFramer in Table 2. By contrast, ReciNet combines competitive per-epoch and inference efficiency with a significantly reduced epoch budget. In the multi-task setting, ReciNet-MT predicts five properties simultaneously with less than twice the training time of single-task ReciNet, while also using fewer parameters than single-task eComFormer. Notably, PotNet reduces its training time by precomputing potential summations, which dominate its inference time

## 5.5 When Does Reciprocal Space Modeling Help?

A natural question is whether reciprocal space modeling provides advantages specifically when long-range interactions dominate. To investigate this, we classify materials in JARVIS-DFT by ionic character using Pauling electronegativity differences ($\Delta X$) between constituent elements. Materials with larger $\Delta X$ exhibit stronger charge transfer and long-range Coulombic interactions that extend well beyond nearest-neighbor coordination shells. Notably, 84.0% of JARVIS compounds exhibit ionic ($\Delta X > 1.7$) or polar ($\Delta X \in [0.5, 1.7]$) character. A detailed analysis and splitting result is provided in Appendix B.9.

We further evaluate representative methods on two subsets of the JARVIS formation energy test set: ionic materials ($\Delta X > 1.7$; 18,539 samples) where long-range Coulombic interactions are dominant, and covalent materials ($\Delta X < 0.5$; 8,137 samples) where bonding is predominantly local. As shown in Table 6, ReciNet achieves a 34% MAE reduction over CGCNN and a 19% reduction over Matformer on the ionic subset, compared to less than 1.3% improvement on the covalent subset. This $35\times$ ratio in relative improvement directly demonstrates that reciprocal space modeling provides its largest gains precisely where long-range electrostatic interactions dominate, validating the physical motivation of our approach.

Table 6: Performance (MAE↓) on ionic vs. covalent subsets of JARVIS formation energy.

| Method | Ionic (18,539) | Covalent (8,137) | Ratio |
|---|---|---|---|
| CGCNN (local) | 48.7 meV/atom | 52.0 meV/atom | – |
| Matformer (global attention) | 39.4 meV/atom | 52.2 meV/atom | – |
| ReciNet (reciprocal) | **32.0 meV/atom** | **51.5 meV/atom** | – |
| *Improvement vs CGCNN* | **34%** | **0.96%** | **35×** |
| *Improvement vs Matformer* | **19%** | **1.3%** | **14×** |

## 6 Conclusion, Limitations, and Future Work

In this work, we introduce ReciNet, a novel model designed to capture both reciprocal space-based long-range and geometric short-range information for crystal property prediction. Empirically, ReciNet achieves substantial performance improvements on three widely used crystal benchmarks, while its multi-task variant with mixture-of-experts (MoE) demonstrates promising results in multi-property prediction. There are several limitations for future enhancement. Firstly, our short-range modeling with geometric GNNs, while effective, could be improved by incorporating recent advancements in Transformer-based models to enhance material representations. Secondly, our work in multi-property prediction represents an initial step that warrants further exploration. Thirdly, extending reciprocal-space modeling to real-world materials where infinite periodicity is less applicable remains an interesting direction for future work. Overall, we encourage the adoption of ReciNet across diverse material systems as a foundational building block.

**Acknowledgments**

Jianan Nie and Peng Gao were supported in part by the National Science Foundation Grant No. 2442171 and the Google Academic Research Award (GARA). The works of Peiyao Xiao and Kaiyi Ji were generously supported by the NSF Career Award under Grant No. 2442418, NSF grants CCF-2311274 and ECCS-2326592. Any opinions, findings, and conclusions made in this paper are those of the authors and do not necessarily reflect the views of the funding agencies.

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

# A Reciprocal Space Concepts

## A.1 Relations Between Real and Reciprocal Lattice Vectors

In materials science, crystals exhibit translational symmetry, where atoms are periodically arranged in 3D space, forming a direct lattice in real space. Each lattice point corresponds to a repeating unit in the crystal structure, described by primitive lattice vectors $\boldsymbol{\ell}_1, \boldsymbol{\ell}_2, \boldsymbol{\ell}_3$. Reciprocal space, on the other hand, represents the spatial frequencies associated with this periodic arrangement. It forms a dual lattice, mathematically related to the real lattice through reciprocal lattice vectors $\boldsymbol{b}_1, \boldsymbol{b}_2, \boldsymbol{b}_3$. $V$ is the volume of the parallelepiped spanned by the three primitive translation vectors of the original Bravais lattice. It remains consistent under periodic permutations of the indices (Economou, 2010).

Volume and vectors are defined as:

$$V = \boldsymbol{\ell}_1 \cdot (\boldsymbol{\ell}_2 \times \boldsymbol{\ell}_3) \tag{11}$$

$$\boldsymbol{b}_1 = 2\pi \cdot \frac{\boldsymbol{\ell}_2 \times \boldsymbol{\ell}_3}{V} \tag{12}$$

$$\boldsymbol{b}_2 = 2\pi \cdot \frac{\boldsymbol{\ell}_3 \times \boldsymbol{\ell}_1}{V} \tag{13}$$

$$\boldsymbol{b}_3 = 2\pi \cdot \frac{\boldsymbol{\ell}_1 \times \boldsymbol{\ell}_2}{V} \tag{14}$$

These vectors satisfy the reciprocal relationship:

$$\boldsymbol{\ell}_i \cdot \boldsymbol{b}_j = 2\pi \delta_{ij}. \tag{15}$$

The symbol $\delta_{ij}$ represents the Kronecker delta, which is a mathematical function defined as:

$$\delta_{ij} = \begin{cases} 1, & \text{if } i = j \\ 0, & \text{if } i \neq j \end{cases} \tag{16}$$

This allows us to define a periodic function naturally in reciprocal space, where vectors $\boldsymbol{k} \in \Lambda$ can be expanded in terms of these reciprocal basis vectors (Ashcroft & Mermin, 2001).

The 3D spatial frequencies must be integer combinations of *three* spatial basis frequencies $\boldsymbol{b}_1, \boldsymbol{b}_2, \boldsymbol{b}_3 \in \mathbb{R}^{3 \times 3}$ spanning the *reciprocal lattice*:

$$\Lambda = \{n_1' \boldsymbol{b}_1 + n_2' \boldsymbol{b}_2 + n_3' \boldsymbol{b}_3 | n_1', n_2', n_3' \in \mathbb{Z}\}. \tag{17}$$

## A.2 Diffraction Methods

Reciprocal space is fundamental to diffraction techniques such as X-ray diffraction (XRD) and electron diffraction. XRD is a widely used technique in crystallography that examines the constructive interference of X-rays scattered by periodic atomic planes, providing detailed insights into crystal structures (Ladd et al., 1977). Similarly, in electron diffraction, the elastic scattering of electrons by atoms produces diffraction patterns that reveal structural information (Egerton et al., 2005). The scattering intensity is modeled using the structure factor:

$$S(\mathbf{G}) = \sum_j f_j e^{-i\mathbf{G} \cdot \mathbf{r}_j} \tag{18}$$

where $\mathbf{G}$ is the reciprocal lattice vector. $f_j$ represents the atomic form factor in X-ray diffraction, and the electron scattering factor in electron diffraction separately. Diffraction patterns are governed by Bragg's law:

$$n\lambda = 2d \sin \theta, \tag{19}$$

relating lattice spacings ($d$) to diffraction angles ($\theta$) and wavelength ($\lambda$) (Cullity & Smoluchowski, 1957). Notably, this condition applies to both XRD and ED, although electron diffraction often requires consideration of dynamical scattering effects due to stronger interactions between electrons and matter (James, 1963).

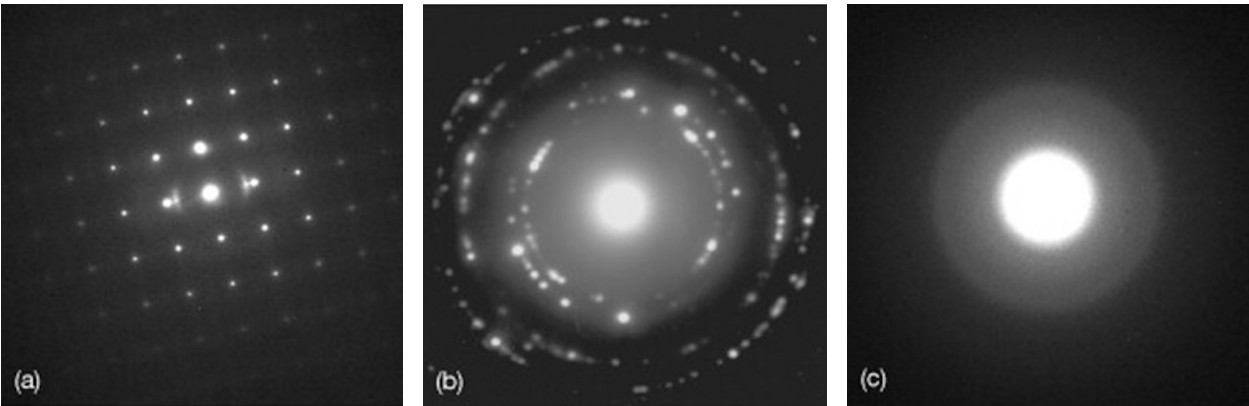

Figure 3: Example of electron diffraction patterns from Poeppelmeier (2023).

Electron diffraction is widely used in transmission electron microscopy (TEM) to analyze materials. For instance, single crystals produce discrete spot patterns, polycrystals form concentric rings, and amorphous materials show diffuse rings (Egerton et al., 2005). These patterns enable crystallinity and orientation analysis, with applications in semiconductors, metals, and oxides. Figure 3 illustrates a typical diffraction pattern. By satisfying the Bragg condition, elastically scattered electrons produce high-intensity spots, with the angular relationship between transmitted and diffracted beams revealing the crystal structure (Poeppelmeier, 2023).

### A.3 Electronic Band

Reciprocal space plays a crucial role in determining electronic and vibrational properties. Specifically, Bloch's theorem relates the electron wavefunction $\psi(\mathbf{r})$ to the wave vector $\boldsymbol{k}$ as:

$$\psi(\mathbf{r}) = e^{i\boldsymbol{k}\cdot\mathbf{r}}u(\mathbf{r}) \tag{20}$$

where $\psi(\mathbf{r})$ is the electronic wavefunction, $\boldsymbol{k}$ is the wave vector in reciprocal space. This theorem facilitates the study of electronic band structures, enabling the prediction of properties like band gaps and carrier mobility (Ashcroft et al., 1978).

$u(\mathbf{r})$ is a function periodic with the lattice, and r denotes position:

$$u(\mathbf{r} + \mathbf{R}) = u(\mathbf{r}) \tag{21}$$

with $\mathbf{R}$ being a lattice translation vector. This decomposition separates the plane wave component, representing long-range propagation, and the periodic modulation, capturing local interactions within the unit cell.

The energy eigenvalues, plotted as a function of $\boldsymbol{k}$ within the first Brillouin zone (Ashcroft & Mermin, 2001), form a band structure that determines key material properties, including band gaps which distinguish insulators, semiconductors, and metals, effective mass and carrier mobility which influence electronic conductivity, and optical absorption spectra and dielectric constants which govern photon-electron interactions and optoelectronic performance (Ziman, 2001).

## B  Experimental Details

### B.1  Dataset

We provide more details for the JARVIS, Materials Project, and MatBench datasets in this section.

Table 7: Dataset sizes vary for different properties.

| Dataset | Task | Total |
|---------|------|-------|
| JARVIS-DFT | Formation Energy | 55,722 |
| | Bandgap (OPT) | 55,722 |
| | Total Energy | 55,722 |
| | Bandgap (MBJ) | 18,171 |
| | $E_{\text{hull}}$ | 55,370 |
| Materials Project | Formation Energy | 69,239 |
| | Band Gap | 69,239 |
| | Bulk Moduli | 5,450 |
| | Shear Moduli | 5,450 |
| MatBench | e_form | 132,752 |
| | jdft2d | 636 |

**JARVIS dataset.** The JARVIS dataset, proposed by Choudhary et al. (2020), contains 55,723 materials and serves as a benchmark for crystal property prediction. Following the experimental protocols of Yan et al. (2024), we evaluate our methods on five regression tasks: formation energy, total energy, bandgap (using both OPT and MBJ methods), and energy above the convex hull (Ehull). The training, validation, and testing splits for formation energy, total energy, and OPT bandgap consist of 44,578, 5,572, and 5,572 crystals, respectively. The splits for Ehull include 44,296, 5,537, and 5,537 crystals, while the MBJ bandgap tasks consist of 14,537, 1,817, and 1,817 crystals. The MBJ functional is considered more accurate for bandgap calculations compared to OPT, and both are utilized in this study. Notably, 18,865 of the dataset's crystal structures have been experimentally observed, adding robustness to its use in predictive modeling.

**The Materials Project dataset.** The Materials Project dataset, introduced by Chen et al. (2019), is a comprehensive collection of 69,239 materials, widely utilized in crystal property prediction studies. We adopt the data splits in Yan et al. (2024) to ensure consistency and fair comparisons with prior methods. Specifically, for formation energy and bandgap prediction tasks, the dataset comprises 60,000, 5,000, and 4,239 crystals for training, validation, and testing, respectively. For bulk modulus and shear modulus predictions, the splits include 4,664, 393, and 393 crystals. Notably, one validation sample in the shear modulus task is excluded due to a negative GPa value, reflecting an underlying unstable or metastable crystal structure. Among the included crystals, 30,084 have been experimentally observed, further highlighting the dataset's reliability for studying material properties.

**MatBench dataset.** The Matbench dataset (Dunn et al., 2020) is a benchmark suite designed for evaluating machine learning models on materials science tasks, offering standardized splits and diverse property predictions. In this work, we focus on two representative regression tasks from Matbench: the large-scale formation energy prediction task (e_form) comprising 132,752 crystal structures, and the smaller-scale 2D exfoliation energy task (jdft2d) with 636 samples. These tasks assess the model's ability to generalize across both abundant and limited data regimes.

### B.2 Settings of Model Embeddings

Node feature is embedded into a CGCNN (Xie & Grossman, 2018) embedding vector of length 92 based on atomic number, and then mapped to a 256-dimensional initial short-range node feature $h_{\text{local}}^0$ by a linear transformation. The initial global node feature $h_{\text{global}}^0$ is obtained by a linear transformation on $h_{\text{local}}^0$ and follows an activation function. The edge feature $||\mathbf{p}_i - \mathbf{p}_j||_2$ is inversely scaled to $-0.75/||\mathbf{p}_i - \mathbf{p}_j||_2$ based on Lin et al. (2023), and then expanded into a 256-dimensional vector by RBF kernels with 256 center values from -4.0 to 4.0; after that, we transfer the 256-dimensional vector to initial edge feature $v_{ij}^e$ through a linear transformation followed by a SoftPlus activation.

## B.3 Geometric GNN Information

Building on PotNet (Lin et al., 2023), we adopt its base model as the local graph component, without using its additional computed infinite summation of interatomic potentials. The local module performs iterative message passing over the constructed graph using a sequence of graph convolutional layers. Each layer updates node features $h$ by aggregating information from neighboring nodes and edge attributes. At layer $l$, messages are constructed by concatenating features of neighboring nodes and edge attributes:

$$\mathbf{z}_{ij}^{(l)} = [h_i^{(l)} \| h_j^{(l)} \| v_{ij}^e] \in \mathbb{R}^{3d}, \tag{22}$$

where $h_i^{(l)}$ and $h_j^{(l)}$ are node embeddings of nodes $i$ and $j$, respectively, $d_{ij}$ is the edge attribute, and $\|$ denotes concatenation. These concatenated features are processed by a multi-layer perceptron to compute attention coefficients that regulate the influence of neighboring nodes:

$$\beta_{ij}^{(l)} = \text{Sigmoid}\left(\text{BN}\left(\text{MLP}\left(\mathbf{z}_{ij}^{(l)}\right)\right)\right) \in \mathbb{R}^d, \tag{23}$$

where $\beta_{ij}^{(l)}$ denotes the local attention coefficient. Messages are then aggregated from neighbors using a weighted sum:

$$\tilde{h}_i^{(l)} = \text{Aggregation}\left(\beta_{ij}^{(l)} \odot \text{MLP}\left(\mathbf{z}_{ij}^{(l)}\right)\right) \in \mathbb{R}^d, \tag{24}$$

with $\odot$ as the Hadamard product and the aggregation operator performing sum or mean pooling. The updated embeddings are computed via residual connections with batch normalization and activation:

$$h_i^{(l+1)} = \text{ReLU}\left(h_i^{(l)} + \text{BN}\left(\tilde{h}_i^{(l)}\right)\right) \in \mathbb{R}^d. \tag{25}$$

## B.4 Hyperparameter Settings of ReciNet on Different Tasks

In this subsection, we share the detailed hyperparameter settings of ReciNet and ReciNet-MT for different tasks in JARVIS, the Materials Project, and MatBench crystal datasets. We use a different number of blocks for different tasks, and higher performance is expected if hyperparameters are tuned specifically for each task.

Table 8: Model settings of ReciNet for the JARVIS dataset.

|  | Num. blocks | Total epochs | Learning rate | Num. neighbors |
|---|---|---|---|---|
| formation energy | 4 | 500 | 0.0008 | 16 |
| band gap (OPT) | 4 | 500 | 0.0006 | 16 |
| band gap (MBJ) | 3 | 500 | 0.001 | 16 |
| total energy | 4 | 500 | 0.0008 | 16 |
| Ehull | 5 | 500 | 0.0006 | 16 |

Table 9: Model settings of ReciNet for the Materials Project dataset.

|  | Num. blocks | Total epochs | Learning rate | Num. neighbors |
|---|---|---|---|---|
| formation energy | 3 | 500 | 0.0008 | 16 |
| band gap | 3 | 500 | 0.0008 | 16 |
| bulk moduli | 3 | 500 | 0.001 | 16 |
| shear moduli | 3 | 300 | 0.001 | 16 |

**JARVIS**. We show the model settings of ReciNet on the JARVIS dataset in Table 8. We train both models using the Adam (Kingma, 2014) optimizer with weight decay of 1e–5, Onecycle scheduler (Smith & Topin, 2019), and for a duration of 500 training epochs. The batch size is standardized at 64, and the models are trained using the L1 loss function. The effectiveness of the model is quantitatively measured using the mean

Table 10: Model settings of ReciNet-MT for JARVIS and the Materials Project datasets.

| | # blocks | Total epochs | Learning rate | # neighbors |
|---|---|---|---|---|
| JARVIS | 3 | 500 | 0.0008 | 16 |
| Materials Project-(form. energy+band gap) | 3 | 500 | 0.0008 | 16 |
| Materials Project-(bulk+shear moduli) | 3 | 300 | 0.001 | 16 |

Table 11: Model settings of ReciNet for MatBench.

| | Num. blocks | Total epochs | Learning rate | Num. neighbors |
|---|---|---|---|---|
| e_form | 5 | 500 | 0.0004 | 16 |
| jdft2d | 3 | 500 | 0.001 | 16 |

absolute error (MAE). The number of neighbors indicates the $k$-th nearest distance we use as the radius for node $i$.

**The Materials Project**. We show the model settings of ReciNet on the Materials Project dataset in Table 9. We train both models using the Adam (Kingma, 2014) optimizer with weight decay of 1e–5, and the Onecycle scheduler (Smith & Topin, 2019). The batch size is standardized at 64, and the models are trained using L1 loss function. The effectiveness of the model is quantitatively measured using the mean absolute error (MAE). The number of training epochs for shear moduli on ReciNet is 300, while for the other three properties it is 500.

**MatBench**. We show the model settings of ReciNet on the MatBench dataset in Table 11. All models are trained for 500 epochs using the Adam (Kingma, 2014) optimizer with MAE loss. The effectiveness of the models is quantitatively measured using the mean absolute error (MAE) and root mean square error (RMSE).

### B.5 Additional Ablation Studies

Table 12: Ablation study of ReciNet with varying model depth, reported in terms of test MAE on the JARVIS dataset. The numbers 3, 4, and 5 denote the number of ReciNet blocks used. The best results are shown in **bold**.

| Method | Form. Energy | Bandgap(OPT) | $E_{\text{total}}$ | Bandgap(MBJ) | $E_{\text{hull}}$ |
|---|---|---|---|---|---|
| | meV/atom | eV | meV/atom | eV | meV |
| ReciNet(3) | 27.8 | 0.127 | 27.2 | 0.242 | 52.9 |
| ReciNet(4) | **27.0** | **0.126** | **26.8** | **0.239** | 48.0 |
| ReciNet(5) | 27.7 | 0.127 | 27.2 | 0.240 | **43.0** |

We further investigate the effect of model depth by varying the number of ReciNet blocks from 3 to 5. As shown in Table 12, increasing the number of blocks from 3 to 4 consistently improves performance across all tasks. Performance on $E_{\text{hull}}$ continues to improve with five blocks, suggesting that deeper message passing may help capture the subtle energy differences relevant to thermodynamic stability. However, for other properties on the JARVIS dataset, adding a fifth block offers no additional benefit and leads to slight performance degradation. This suggests that while increased depth enhances representational capacity, it may also introduce overfitting or optimization difficulties, especially when training data is limited. To balance predictive accuracy and computational cost, we adopt three or four ReciNet blocks as the default configuration.

Table 13: Comparison of ReciNet with machine learning models incorporating reciprocal space. For EwaldMP, we use $N_x = N_y = N_z = 1$ for simplicity. For Crystalformer, dual space means the combined use of real space and reciprocal space attention. The best results are highlighted in **bold**, and the second-best results are indicated with underlines.

| Method | Form. Energy | Bandgap(OPT) | $E_{total}$ | Bandgap(MBJ) | $E_{hull}$ |
|---|---|---|---|---|---|
| | meV/atom | eV | meV/atom | eV | meV |
| EwaldMP (Kosmala et al., 2023) | 31.0 | 0.132 | 31.6 | 0.26 | 57 |
| PotNet (Lin et al., 2023) | 29.4 | 0.127 | 32 | 0.27 | 55 |
| Crystalformer (Taniai et al., 2024) (dual space) | 34.3 | 0.126 | 35.3 | 0.283 | **28.4** |
| **ReciNet** | **27.0** | **0.126** | **26.8** | **0.24** | 43 |

## B.6 Additional Experimental Comparisons with Reciprocal Space-Based Methods

**EwaldMP.** We have provided additional benchmarking results on the JARVIS dataset, where both ReciNet and EwaldMP (Kosmala et al., 2023) are evaluated. As shown in the Table 13, ReciNet consistently outperforms EwaldMP across all five target properties, demonstrating the effectiveness of our approach beyond EwaldMP for bulk crystalline materials. EwaldMP integrated structure factor embeddings directly into GNN message passing, enabling feedback to node embeddings. However, it requires extensive hyperparameter tuning for grid sizes $(N_x, N_y, N_z)$ to define the supercell, which can disrupt the symmetry of the unit cell, alter its space group, and substantially increase computational costs, particularly for systems with complex symmetries. While EwaldMP incorporates reciprocal space modeling, their approach is tailored to molecular systems and evaluated on semi-periodic surface structures from the OC20 dataset, which includes relaxed small-molecule adsorbates (e.g., CO, $H_2O$, $NH_3$) on metal surfaces (e.g., Pt, Cu, Ni). These systems lack full 3D periodicity and space group symmetries typical of bulk crystals. As a result, the study does not assess reciprocal space modeling in fully periodic crystalline materials such as perovskites (e.g., $SrTiO_3$) or semiconductors (e.g., GaAs), which are typical and important materials. In contrast, our method is explicitly designed for bulk crystals and directly encodes their symmetry and periodicity.

**PotNet.** PotNet (Lin et al., 2023) explicitly models long-range interactions by using the Ewald summation technique, which calculates the long-range portion of the potential in reciprocal space. However, these potentials are pre-computed and fed into the GNN as fixed, non-trainable edge features. That says, PotNet does not perform learning or message passing in reciprocal space. This limits the model's ability to adaptively learn complex interactions from different crystalline materials. Also, PotNet incurs an additional $\mathcal{O}(n^2)$ computational cost due to pre-calculation (which is shown in Table 5 where the inference time of PotNet is dominated by the precomputation of potential summations). This makes it inefficient for complex material systems with large $n$, where $n$ is the number of atoms in the unit cell. In contrast, ReciNet integrates the reciprocal space transformation directly into the network's forward pass, applying fully learnable filters to the Fourier-space representations in an end-to-end manner. The comparison with PotNet is shown in Table 13 on the JARVIS dataset, which validates these advantages. ReciNet achieves lower MAE values for all five tasks than PotNet, especially lower MAE for formation energy (27.0 vs. 29.4), $E_{hull}$ (43 vs. 55), and total energy (26.8 vs 32). Together, these demonstrate the superiority of ReciNet compared to PotNet.

**Crystalformer.** Separately, Crystalformer (Taniai et al., 2024) introduces reciprocal space via fixed analytical Fourier-Gaussian formulations within a Transformer-style architecture in their appendix. However, their reciprocal space component is non-trainable and presented only as an optional variant in Appendix J. Our proposed ReciNet instead employs fully learnable reciprocal space filters, trained end-to-end to capture long-range interactions beyond the capacity of fixed-function decays. While both models incorporate reciprocal space to enhance long-range interaction modeling, ReciNet consistently demonstrates stronger performance. As shown in Table 13, ReciNet outperforms Crystalformer (dual space variant from Appendix J in Crystalformer) in the JARVIS dataset. Specifically, ReciNet reduces the MAE by 21.3% for formation energy (0.0270 vs. 0.0343), 24.1% for total energy (0.0268 vs. 0.0353), and 15.2% for Bandgap(MBJ) (0.24

Table 14: Comparison of MTL on JARVIS in terms of test MAE. N/A denotes that the property is not involved in the prediction.

| Method | Form. Energy | Bandgap(OPT) | $E_{\text{total}}$ | Bandgap(MBJ) | $E_{\text{hull}}$ |
|---|---|---|---|---|---|
| | meV/atom | eV | meV/atom | eV | meV |
| ReciNet-MT | **35.0** | **0.122** | **36.0** | **0.21** | **69.2** |
| ReciNet-LS (3 tasks) | 37.0 | 0.127 | 37.3 | N/A | N/A |
| ReciNet-LS (5 tasks) | 49.9 | 0.136 | 53.0 | 0.24 | 92.1 |

vs. 0.283). These substantial improvements highlight the effectiveness of our learnable reciprocal space design in capturing complex, long-range dependencies in crystalline materials.

## B.7 Training Time Scale Efficiency Analysis

Beyond the superior modeling capacity for crystalline materials, our ReciNet is faster and more efficient than previous works. To demonstrate the efficiency of ReciNet, we compare ReciNet and its multi-task variant, ReciNet-MT, with iComFormer, PotNet, Matformer, Crystalformer, and CrystalFramer. Evaluation is in terms of training time per epoch on the task of JARVIS formation energy prediction, the number of training epochs, total training time, and average inference time per material. From Table 5, ReciNet demonstrates superior computational efficiency. Specifically, ReciNet outperforms PotNet by 1.2×, Matformer by 1.7×, iComFormer by 1.7×, and CrystalFramer by 2.11× speedup. Finally, the time for ReciNet-MT predicting 5 properties does not even double compared to single property prediction. These results highlight the effectiveness of ReciNet in reducing computational overhead while maintaining performance in prediction, as previously mentioned.

## B.8 ReciprocalBlock as a Plug-and-Play Module

To evaluate the generalizability of the ReciprocalBlock beyond the ReciNet architecture, we integrate it as a plug-and-play component into two existing backbones: CGCNN and Matformer. For each backbone, we append ReciprocalBlock layers that operate in parallel with the original message-passing layers, following the same integration scheme described in Section 4, with same hyperparameters settings.

As shown in Table 15, integrating the ReciprocalBlock yields consistent improvements across all five JARVIS properties for both backbones. For CGCNN, the addition reduces formation energy MAE from 63 meV/atom to 36.2 meV/atom (42.5% improvement), demonstrating that even a simple local GNN benefits substantially from reciprocal space information. For Matformer, increasing the ReciprocalBlock depth from 2 to 3 layers further reduces errors (e.g., formation energy: 32.5 meV/atom→ 29.9 meV/atom→ 28.9 meV/atom), indicating that deeper reciprocal modeling captures additional long-range structure. These results confirm that the ReciprocalBlock is a generally effective module for capturing long-range dependencies, capable of enhancing different backbone architectures.

Table 15: Additional results of adding ReciprocalBlock with CGCNN and Matformer performance on the JARVIS dataset.

| Method | Form. Energy | Bandgap(OPT) | $E_{\text{total}}$ | Bandgap(MBJ) | $E_{\text{hull}}$ |
|---|---|---|---|---|---|
| | meV/atom | eV | meV/atom | eV | meV |
| CGCNN | 63 | 0.20 | 78 | 0.41 | 170 |
| CGCNN with Reciprocal | 36.2 | 0.158 | 51.4 | 0.328 | 65 |
| Matformer | 32.5 | 0.137 | 35 | 0.30 | 64 |
| Matformer with Reciprocal (2 layer) | 29.9 | 0.130 | 29.3 | 0.260 | 49 |
| Matformer with Reciprocal (3 layer) | 28.9 | 0.127 | 28.2 | 0.245 | 46 |

### B.9   JARVIS-DFT Dataset Composition by Ionic Character

We classified all 55,723 materials in JARVIS-DFT by ionic character using Pauling electronegativity differences ($\Delta X$), as shown in Table 16. Materials with larger $\Delta X$ exhibit stronger charge transfer and ionic character, leading to long-range interactions that extend well beyond nearest neighbors. For single-element materials, we classify them as elemental.

Specifically, 84.0% of JARVIS compounds exhibit ionic or polar character where long-range interactions extend beyond the local coordination shells and contribute substantially to crystal properties, especially formation energy and total energy.

To further analyse, we report the performance of different representative methods, including CGCNN, Matformer, and our ReciNet, on the splitted subset of the JARVIS formation energy dataset. Specifically, we test on two subsets, where long-range interactions are critical in ionic: $\Delta X > 1.7$), versus negligible in covalent: $\Delta X < 0.5$). The results are shown in Table 6. We show that ionic materials have $35\times$ larger improvement than covalent (34% vs. 0.96%), directly demonstrating that reciprocal space modeling provides advantages specifically when long-range interactions dominate.

Table 16: Ionic character classification of JARVIS-DFT materials by electronegativity difference.

| Material Class | Count | Percentage | Long-Range |
|---|---|---|---|
| Ionic ($\Delta X > 1.7$) | 18,539 | 33.3% | High |
| Polar ($\Delta X = 0.5$–1.7) | 28,246 | 50.7% | Moderate |
| Covalent ($\Delta X < 0.5$) | 8,137 | 14.6% | Low |
| Element | 777 | 1.4% | N/A |

### B.10   Deviation Analysis

To assess the stability of ReciNet across random initializations, we report the mean and standard deviation over 5 independent runs on the JARVIS and Materials Project datasets in Table 17 and Table 18, respectively. The consistently low standard deviations across all tasks indicate that ReciNet exhibits stable training behavior and that the performance gains reported in the main text are not sensitive to seed selection. This complements the multi-run results reported for MatBench in Table 3.

Table 17: Mean $\pm$ standard deviation for all tasks in JARVIS dataset.

| | Form. Energy | Bandgap(OPT) | $E_{\text{total}}$ | Bandgap(MBJ) | $E_{\text{hull}}$ |
|---|---|---|---|---|---|
| | meV/atom | eV | meV/atom | eV | meV |
| ReciNet | 27.0$\pm$0.04 | 0.126$\pm$0.0002 | 26.8$\pm$0.04 | 0.24$\pm$0.01 | 43$\pm$0.03 |

Table 18: Mean $\pm$ standard deviation for all tasks in Materials Project dataset.

| | Formation Energy | Band Gap | Bulk Moduli | Shear Moduli |
|---|---|---|---|---|
| | meV/atom | eV | log(GPa) | log(GPa) |
| ReciNet | 17.07$\pm$0.02 | 0.189$\pm$0.003 | 0.0328$\pm$0.0005 | 0.0628$\pm$0.0002 |

## C   Multi-task Learning

### C.1   Motivation

While our novel usage of reciprocal space representations effectively captures long-range interactions and global periodicity, it still adheres to the traditional approach of predicting properties individually (Schütt et al., 2017; Choudhary & DeCost, 2021; Yan et al., 2024). This approach becomes computationally and memory inefficient when repeatedly applied to multi-property predictions. To address this limitation, Multi-Task Learning (MTL) offers a promising solution by enabling the prediction of multiple properties within a single model (Evgeniou & Pontil, 2004). Additionally, MTL can leverage positive transfer, where knowledge and representations learned from one task can benefit other tasks. This advantage is particularly relevant for materials, as similar properties (e.g., OPT bandgap and MBJ bandgap) are often highly correlated and share similar structural features. However, to our knowledge, few studies have specifically investigated multi-property prediction for materials.

### C.2   Related Works on Multi-property prediction

While multi-property prediction in materials remains largely unexplored, we summarize several advances in molecular domains. Liu et al. (2022) introduced the MTL concept in molecular property prediction in the early stage. Their approach involves constructing a sparse dataset and relying on prior knowledge of property relations before training. Christofidellis et al. (2023) proposed a language model that bridges natural and chemical languages to perform multi-task learning, such as chemical reaction prediction and retrosynthesis. However, their work is not specifically focused on multi-property prediction. Additionally, Beaini et al. (2023) developed a foundation model and a large-scale molecular dataset tailored for multi-task learning. Nevertheless, there is currently no comparably large material dataset to support a similar effort. In another approach, Ren et al. (2024) leveraged physical laws after task decoders to address data heterogeneity across different properties, improving prediction consistency. In contrast, we integrate the Mixture of Experts (MoE) between encoders and decoders, allowing each task to obtain customized structural features before property prediction.

### C.3   MTL and MoE

Multi-Task Learning (MTL) typically employs a shared backbone network to extract common features, followed by task-specific output heads (decoders) that specialize in individual predictions (Xiao et al., 2024). In this context, the Mixture of Experts (MoE) framework offers a flexible and scalable approach to capture both shared and task-specific features by leveraging task-relevant experts. MoE has gained significant attention in multi-task scenarios across computer vision and natural language processing due to its adaptability and efficiency (Riquelme et al., 2021). An MoE layer consists of a set of expert networks, denoted as $E_i, \forall i \in [1, N]$, where $N$ is the number of experts, along with a routing network $R$. The output of an MoE layer is the weighted sum of the output $E_n(x)$ from every expert, where weights $G_n(x)$ are calculated by the routing network $R$ and $x$ represents the model input. Formally, the output of a MoE layer is given by

$$y = \sum_{i=1}^{N} G_n(x)E_n(x). \tag{26}$$

The routing network utilizes a noisy Top-$K$ selection mechanism (Shazeer et al., 2017) to model the probabilities of each expert and select the top $K$ candidates. This process is defined as:

$$G(x) = \text{TopK}(\text{Softmax}(xW + \mathcal{N}(0, 1)\text{Softplus}(xW_{\text{noise}}))), \tag{27}$$

where $W$ and $W_{\text{noise}}$ are router model parameters, $\mathcal{N}(0, 1)$ is a standard normal distribution, and $\text{Softmax}(\cdot)$ and $\text{Softplus}(\cdot)$ are activation functions. Besides, the $\text{TopK}(\cdot)$ function retains only the $K$ largest values, setting all others to zero.

Table 19: Comparison of MTL on The Materials Project in terms of test MAE.

| Method | Formation Energy | Band Gap | Bulk Moduli | Shear Moduli |
|---|---|---|---|---|
| | meV/atom | eV | log(GPa) | log(GPa) |
| ReciNet-MT | **24.50** | **0.218** | **0.0391** | **0.0668** |
| ReciNet-LS | 39.10 | 0.221 | 0.0404 | 0.0713 |

### C.4 Multi-Property Prediction Decoder

For multi-property prediction, we employ ReciNet-MT, a multi-task learning variant of ReciNet in the decoder that incorporates mixture-of-experts (MoE) layers followed by task-specific fully connected heads. We assume that the aggregated node features from the final ReciNet block encapsulate both short-range and long-range information of materials, which can be shared among all property prediction tasks. However, different properties require customized features for individual task heads. Therefore, experts are expected to learn distinct features, while the routing network can determine the optimal expert combination. Unlike the post-processing of the predicted property values after decoders with physical laws (Ren et al., 2024), this approach offers more flexibility. Meanwhile, similar properties could have a large expert selection overlap, such that the knowledge can be shared among them. Notably, this expert-sharing mechanism could facilitate positive transfer.

### C.5 Detailed Results on Multi-Property Prediction

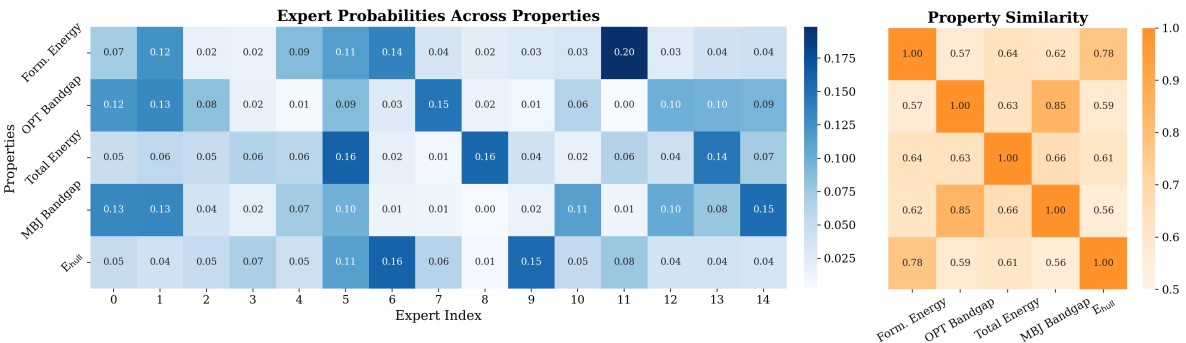

Figure 4: MoE experts' frequencies for each property on the JARVIS dataset and property similarities. The expert selections in the OPT bandgap and MBJ bandgap show the highest similarity, and the positive transfer happens between these two predictions.

In this section, we provide detailed results of using ReciNet-MT on JARVIS and The Materials Project datasets. We compare our method with the most accepted MTL method, linear scalarization (LS), and the results are shown in Figure 4, Table 14, and Table 19.

Performance degradation in MTL compared to STL is a well-known phenomenon in the MTL literature. This issue is commonly attributed to the gradient conflict, which arises when gradients from different tasks (properties) point in conflicting directions or have significantly different magnitudes (Xiao et al., 2024). Therefore, it is realistic to expect a performance drop in both Table 14 and Table 19 for most properties, and this phenomenon becomes severe when the number of properties increases. However, our ReciNet-MT still matches Matformer and ALIGNN on both datasets.

Surprisingly, ReciNet-MT achieves the best performance for two bandgaps, though some degradation happens. Notably, bandgap (MBJ) surpasses the state-of-the-art (SOTA) results of single-task models, while bandgap (OPT) matches the SOTA performance. To evaluate this improvement, we track the selection frequencies of the 15 experts and visualize the expert selection probabilities for each task (see Figure 4, left).

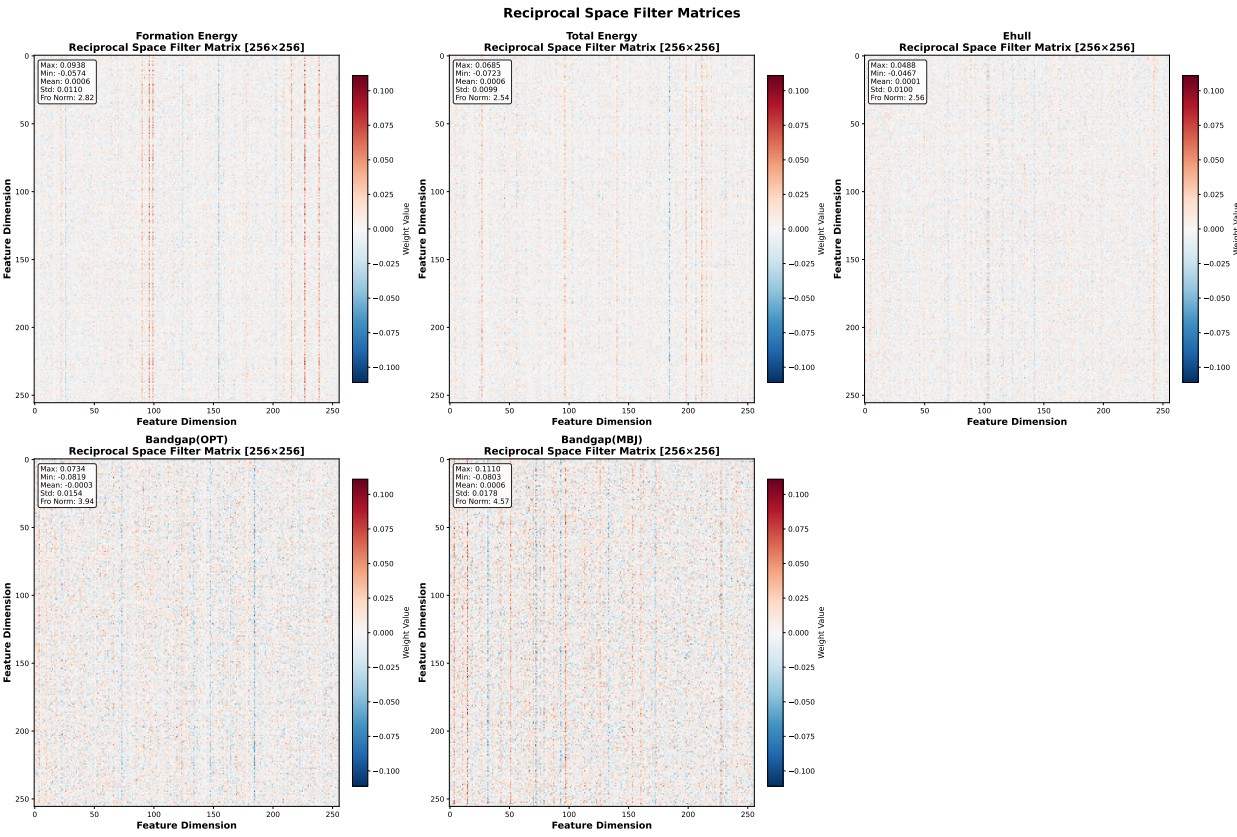

Figure 5: Learned reciprocal space filter matrices for JARVIS dataset, revealing property-specific weighting patterns.

We also compute the cosine similarity among properties via expert selection probabilities (see Figure 4, right). Among the five tasks, two bandgap properties exhibit the highest similarity score of 0.85, indicating a big overlap in the expert selection, which facilitates the positive transfer. This performance gain can be attributed to positive transfer across related properties; specifically, both band gap (OPT) and band gap (MBJ) are governed by the underlying electronic structure, enabling shared representations to benefit both tasks. This finding explains why ReciNet-MT outperforms single-task training for two band gaps.

## D   Physical Interpretability

To provide a physical interpretability for our reciprocal space filter, we provide a comprehensive analysis of the reciprocal space filter in eq. (9), using both direct visualization and Singular Value Decomposition (SVD). Our findings indicate that ReciNet adaptively learns property-specific filters that align with the underlying physics of crystal materials in some way.

### D.1   Methodology

We perform Singular Value Decomposition on the learned Fourier filter matrices $\boldsymbol{W}_{\text{filter}} \in \mathbb{R}^{256 \times 256}$ from trained ReciNet models across five JARVIS-DFT properties. For each property-specific model, we extract the filter weight matrix from the ReciprocalBlock and compute:

$$\boldsymbol{W}_{\text{filter}} = U\Sigma V^T = \sum_{i=1}^{256} \sigma_i u_i v_i^T \tag{28}$$

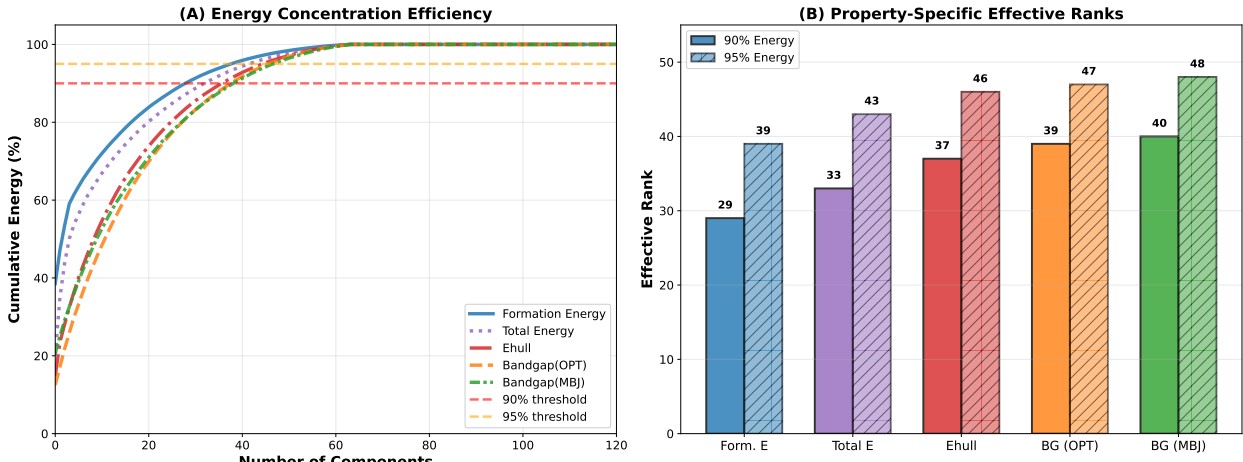

Figure 6: Singular value decomposition analysis of learned reciprocal space filters.

where $\sigma_1 \geq \sigma_2 \geq \cdots \geq \sigma_{256} \geq 0$ are singular values in descending order. $u_i$ and $v_i$ are the left singular vector and the right singular vector, respectively..

**Effective Rank Definition.** The effective rank is defined as:

$$\text{Rank}_{\alpha\%} = \min\left\{ k : \frac{\sum_{i=1}^{k} \sigma_i^2}{\sum_{i=1}^{256} \sigma_i^2} \geq \frac{\alpha}{100} \right\} \tag{29}$$

This quantifies the intrinsic dimensionality of the learned representation. We report both $\text{Rank}_{90\%}$ (primary metric) and $\text{Rank}_{95\%}$ to assess concentration of filter energy.

### D.2 Qualitative Analysis via Filter Visualization

We first visualize the learned reciprocal space filter matrices $\boldsymbol{W}_{\text{filter}}$ across varying properties (see Figure 5). Formation Energy exhibits distinct, sparse stripe patterns, indicating that the model selectively emphasizes specific components while suppressing others. Band Gap (OPT/MBJ), in contrast, displays a dense, uniform weight distribution with minimal sparsity. Ehull and Total Energy display intermediate characteristics.

### D.3 Quantitative Analysis via Singular Value Decomposition

To quantify these observations, we applied Singular Value Decomposition (SVD) to the filter matrices (see Figure 6). We compute the Effective Rank (a widely accepted metric for measuring representational complexity), defined as the number of singular components required to capture 90% of the cumulative energy for the matrix. The results reveal a striking progression in model: Formation Energy (Rank 29) < Total Energy (Rank 33) < E-hull (Rank 37) < Bandgap(OPT) (Rank 39) < Bandgap(MBJ) (Rank 40). This quantification demonstrates that Formation Energy can be modeled with a relatively low-rank, compact representation, whereas Band Gaps require a high-rank, high-capacity representation.

### D.4 Physical Interpretation of Learned Ranks

Importantly, this hierarchy of learned ranks aligns with the underlying physics of crystal properties to some extent. Formation energy relates to specific reciprocal space since it is dominated by long-range Coulomb potentials, which decay as $1/r$ in real space, and the corresponding signal in reciprocal space follows an analytical decay. The model, therefore, requires fewer components to capture these electrostatic sums, resulting in the observed low-rank filter. In contrast, electronic band gaps depend on complex quantum mechanical effects, including orbital overlap, crystal field splitting, and subtle geometric distortions, whose

structural signatures do not reduce to a few dominant Fourier components. Consequently, a high-rank filter is required to capture broader representation to encode the richer structure-property relationships governing electronic properties.

In conclusion, this analysis provides evidence that the ReciprocalBlock obtains an adaptive, physics-aware filter. It tends to allocate compact filters for smooth electrostatic properties and complex filters for intricate electronic properties.

