# OpenReview forum: "ReciNet: Reciprocal Space-Aware Long-Range Modeling for Crystalline Property Prediction"
_TMLR — Accepted by TMLR_

### Review · Reviewer_Gxue · 2026-03-27

**Summary Of Contributions:**

Property prediction from the structure of crystals remains a challenging problem due to local and global dependencies in atomic arrangements. Unlike molecules, crystal structures possess an infinite periodic repetition of atoms in the lattice unit cell. The paper aims to tackle long-range interactions by utilizing the reciprocal space (fourier space) of the lattice vectors. The work proposes ReciNet, a framework for predicting properties from crystal structures using Short-range GNN blocks and the long-range Reciprocal block. Graph embeddings are processed using the CGCNN encoder, followed by short and long range encoding. At its core, the ReciNet uses the Reciprocal block to extract global representations using a weighted exponential transform of reciprocal lattice vectors and fractional coordinates of atoms. Output representations are the sum of reciprocal and inverse fourier-transformed embeddings (using a weighting filter). ReciNet is compatible with both single and multi task settings wherein the latter case, a mixture of experts decoder is utilized to simultaneously predict properties. Authors demonstrate comparisons with pre-existing baselines on Materials Project, JARVIS-DFT and MatBench. ReciNet is found to be suitable in property prediction with long-range Reciprocal blocks capturing global feature information.

**Additional Comments:**

NA

**Audience:**

Yes

**Audience Explanation:**

The paper studies the important problem of crystal structure property prediction from the viewpoint of a systems/engineering perspective. ReciNet focusses on an important application and motivates design decisions which may be further utilized in other domains of pure and applied sciences. The method, and its findings, may be of interest to both the machine learning community as well as the material science/engineering community.

**Claims And Evidence:**

No

**Claims Explanation:**

* **Empirical Evaluation:** The paper conducts comprehensive experiments and comparisons with a range of baselines on standardized benchmarks. However, evaluation of ReciNet and baselines is significantly limited. Comparisons only utilize the MAE (and RMSE) as a metric to evaluate and compare performance. A comparison and analysis of goodness-of-fit/explainability in property prediction is also not provided. Finally, evaluation scheme does not follow a uniform trend wherein comparisons on Materials Project and JARVIS do not provide error differences. With the current evaluation setup, claims of improved performance are not well supported. Can the authors report correlation, calibration and percentage error above a threshold? How does ReciNet evaluate across explainability (R2 and explained variance)? Can all tables have error ranges and number of runs?

* **Ablations:** The paper provides an ablation study but this is limited in scope as it does not evaluate the main components of ReciNet. Can MTL and STL be explicitly compared to observe the role of mixture of experts as a signficant contribution for MTL? How mayn experts are ideal? How do multiple Reciprocal blocks per layer impact prediction/long-range feature information? How does the omission of short-range blocks affect prediction? In my view, the (1) reciprocal block, (2) mixture of experts decoder and (3) short-range blocks should be studied in isolation as these are the main components of ReciNet.

* **Intuition and Knowledge:** The paper provides intuitive insights about the role of reciprocal block in capturing long-range feature information using the fourier space. However, these claims are not validated. Can authors qualitatively (and practically) show that long range information is being exploited using the fourier domain? Can authors also dicsuss the possibility of smoothing and aliasing between reciprocal lattice vectors? Authors could consider adding qualitative results or aseessing the fourier components to validate claims.

**Requested Changes:**

Following are the changes critical for acceptance -

* Empirical Evaluation - A uniform and comprehensive evaluation of ReciNet and baseline methods in experiments. This would include additional metrics (correlation, percentage error, calibration) and goodness-of-fit (R2, explained variance). Standardized error ranges and number of runs will further help validate the statistical signficance.

* Ablations - Studying the role of components of ReciNet in isolation, (1) Reciprocal block, (2) Mixture of experts and (3) short-range blocks. In the case of reciprocal blocks, adding or reducing number of blocks per layer and permuting their placement would further strengthen ablations on top of pre-existing ones.

Following are the changes that would strengthen the quality of work -

* Intuition - Qualtitative results or discussion about the use of reciprocal block in capturing long-range feature information. Authors could evaluate lattice vectors in the fourier space to verify their claims and assess smoothing and aliasing for a deeper analysis.

---

> ### Author Response · Authors · 2026-04-03
>
> **1. Empirical Evaluation**
>
> **We have conducted additional experiments to further validate our results, including error ranges, statistical significance tests, and $R^2$ scores.**
>
> (1) **Error ranges, Number of runs, and Goodness-of-fit.** We conducted multiple-seed experiments (5 random seeds) for ReciNet on the JARVIS and Materials Project datasets. The results below report the mean ± standard deviation and $R^2$  score for all tasks. These will be included in the revised manuscript.
>
> Table: Mean ± standard deviation and $R^2$ score for all tasks in JARVIS dataset
> |Metric|Form. Energy (meV/atom)|Bandgap(OPT) (eV)|Etotal (meV/atom)|Bandgap(MBJ) (eV)|Ehull (meV)|
> |-|-:|-:|---:|-:|-:|
> |MAE|27.0±0.04|0.126±0.0002|26.8±0.04|0.24±0.01|43±0.03|
> |R² score|0.997|0.935|0.999|0.947|0.992|
>
> Table: Mean±standard deviation and $R^2$ score for all tasks in Materials Project dataset
> |Metric|Formation Energy (meV/atom)|Band Gap (eV)|Bulk Moduli (log(GPa))|Shear Moduli (log(GPa))|
> |-|-:|-:|-:|-:|
> |MAE|17.07±0.02|0.189±0.003|0.0328±0.0005|0.0628±0.0002|
> |R² score|0.999|0.957|0.975|0.956|
>
> (2) **Statistical significance.** ReciNet's improvements over the representative strong baseline are statistically significant.
> We conducted paired significance tests on the Materials Project dataset, comparing ReciNet with iComFormer, across all four properties in the table below. The improvements are statistically significant (p $\leq$ 0.01  for most properties). These results confirm that ReciNet's gains are not attributable to random variation.
>
> Table: Paired significance test between ReciNet and iComFormer on the Materials Project dataset
> | |Form. Energy|Bandgap|Bulk Moduli|Shear Moduli|
> |-|-:|-:|-:|-:|
> |p value|0.0002|0.0942|0.0013|0.0088|
>
> (3) **Regarding correlation, percentage error, and calibration.** Our evaluation protocol, using MAE as the primary metric, is standard across the crystal property prediction literature, including CGCNN, ALIGNN, Matformer, PotNet, ComFormer, and CrystalFramer. Metrics such as correlation, percentage error, and calibration are not reported by any of these works. Nonetheless, we have provided $R^2$ scores, error ranges, and statistical significance as supplementary evidence to further validate our results.
>
> **2. Ablations**
>
> **The requested ablation studies on the reciprocal block, short-range block, and block depth are already included in the paper.**
>
> (1) **Reciprocal block and short-range block.** Table 4 (Section 5.2) provides a factorial ablation: ReciNet w/o Reciprocal represents short-range block only, ReciNet represents with reciprocal block. Comparing the two variants shows consistent improvements across all 5 JARVIS properties when the reciprocal block is included.
>
> (2) **Varying block depth.** Table 4 and Table 12 show ablations on varying the number of ReciNet blocks. Additionally, Table 15 (Appendix B.8) demonstrates the ReciprocalBlock as a plug-and-play module on CGCNN and Matformer, confirming architecture-agnostic effectiveness.
>
> (3) **Block placement.** As illustrated in Figure 2(b), the short-range GNN and reciprocal block operate in parallel within each ReciNet block by design, processing local and global information simultaneously. Since the two branches execute concurrently, there is no sequential ordering to permute.
>
> (4) **Mixture-of-experts decoder.** The MoE decoder is not a core architectural contribution but a supplementary extension. Its purpose is to show the potential of multi-property prediction with a single model instead of $K$ models. Therefore, a dedicated ablation study of this component is not required.
>
> **3. ReciprocalBlock Analysis**
>
> **The ReciprocalBlock captures meaningful long-range information, supported by both existing empirical validation and new interpretability analysis.**
>
> (1) **New interpretability analysis of the learned filter (Appendix D).** We conducted a comprehensive physical interpretability analysis of the learned reciprocal space filter $\boldsymbol{W}_\text{filter}$ in Eq. (9), using both direct visualization and Singular Value Decomposition (SVD). **The full analysis is presented in Appendix D of the updated manuscript.** Our analysis shows that the ReciprocalBlock adaptively learns property-specific filters whose structure aligns with the underlying physics of crystal materials, providing direct qualitative evidence that the reciprocal block captures meaningful long-range feature information.
>
> (2) **Existing empirical validation via ionic vs. covalent subset (Section 5.5).** Our manuscript already provides empirical evidence on this point. We classified materials in JARVIS by ionic characters ($\Delta$X), shown in Appendix B.9. We reported the performance of different methods on the split subset of the JARVIS formation energy dataset. The results demonstrate that the reciprocal block provides significant advantages when long-range interactions dominate, and highlight ReciNet's effectiveness in capturing long-range dependencies.

---

### Review · Reviewer_saVj · 2026-04-05

**Summary Of Contributions:**

This paper introduces ReciNet, a crystal property prediction model that combines a local geometric GNN with a reciprocal-space block to capture both short-range interactions and long-range periodic structure. The main idea is physically well motivated: reciprocal space is a natural domain for periodic crystals, and the proposed learnable Fourier-style representation is an appealing way to model long-range effects. The paper also presents a multi-task extension, ReciNet-MT, and evaluates the approach on Materials Project, JARVIS, and MatBench, where it shows strong overall performance, useful ablations, and encouraging efficiency results. Main strengths are the clear motivation, strong empirical coverage, and solid performance across benchmarks. A minor weakness is that some of the strongest claims could be phrased a bit more carefully in places.

**Audience:**

Yes

**Audience Explanation:**

I expect this work to be of interest to researchers in materials machine learning, geometric deep learning, and scientific ML more broadly. The use of reciprocal-space modeling is both physically meaningful and technically interesting, and the paper addresses an important challenge in learning over periodic crystal structures. The ideas are relevant beyond the specific benchmark setting and may inspire further work on structure-aware learning for physical systems.

**Broader Impact Concerns:**

I do not have major broader-impact concerns.

**Claims And Evidence:**

Yes

**Claims Explanation:**

The paper provides convincing evidence through results on several standard benchmarks, ablation studies removing the reciprocal component, comparisons to prior reciprocal-space-related approaches, efficiency analysis, and additional analysis on ionic versus covalent materials. Altogether, the experimental section gives a strong and coherent picture that the reciprocal-space design is useful and practically effective for crystal property prediction. While some margins are modest and uncertainty reporting could be more complete in a few tables, the overall empirical support is clear and sufficient.

**Requested Changes:**

1. **Clarify the main performance claims.** Since ReciNet is not the top-performing method on every reported task, I encourage the authors to revise the wording around “state-of-the-art” and related claims so that it more precisely reflects where the method is best, where it is competitive, and where the gains are more modest. This would make the presentation more balanced with respect to Tables 1–3.

2. **Report uncertainty estimates in the main benchmark tables.** Please include standard deviations or confidence intervals for the main JARVIS and Materials Project results, not only for MatBench. This would be especially helpful in cases where the gap to strong baselines is small, so that readers can better assess the significance and robustness of the reported improvements.

3. **Clarify the fairness of the reciprocal-space baseline comparisons.** In particular, I would appreciate a clearer explanation of the simplified EwaldMP setting used in Table 13, as well as the rationale for excluding Crystalformer from the main JARVIS comparison while still discussing it later in the reciprocal-space and efficiency analyses. A brief discussion of training budget, implementation choices, and overall comparability would strengthen this part of the paper.

4. **Strengthen the support for the symmetry-preservation claim.** The discussion that the proposed reciprocal-space design preserves crystal-native periodic structure is well motivated, but at present it is supported mainly by architectural intuition. This claim would be stronger with a small empirical check showing consistent predictions under equivalent lattice or coordinate representations. Otherwise, I would suggest softening the wording slightly.

5. **Qualify the interpretability discussion more carefully.** The appendix analysis of the learned reciprocal-space filters is interesting and adds value to the paper. However, some of the conclusions currently read more strongly than the evidence directly supports. I suggest presenting this section as suggestive interpretability analysis rather than as definitive physical validation, unless additional evidence is provided.

---

> ### Author Response · Authors · 2026-04-10
>
> **Requested change 1.**
>
> We thank the reviewer for this suggestion. We have revised the manuscript to more precisely characterize our empirical results in the abstract, introduction, and our experimental summary (Section 5.1). We believe this revised manuscript provides a more balanced presentation with respect to Tables 1–3.
>
> **Requested change 2.**
>
> We appreciate this suggestion. In the revised manuscript, we include deviation results for ReciNet on both the JARVIS and Materials Project datasets in Appendix B.10 (Tables 17 and 18). The results suggest that ReciNet training is stable and that the reported results are not sensitive to random seed selection.
>
> **Requested change 3.**
>
> **Explanation of the simplified EwaldMP.** As noted in the Table 13 caption, we adopt $N_x=N_y=N_z=1$ for EwaldMP, which corresponds to the minimal supercell grid size. This configuration is employed because EwaldMP was originally designed for semi-periodic surface structures (OC20), and its grid dimensions $N_x,N_y,N_z$ require per-system hyperparameter tuning.
>
> **Rationale for excluding CrystalFramer.**
> We believe the reviewer is referring to CrystalFramer rather than Crystalformer, as Crystalformer is already included in our main JARVIS comparison (Table 2), reciprocal-space comparison (Table 13), and efficiency analysis (Table 5). To ensure a fair comparison under comparable training budgets, CrystalFramer was excluded from the JARVIS experiments.
> Specifically, from Table 5 in the paper, CrystalFramer requires 2000 epochs with 41.2h total training time, whereas our method only requires 500 epochs with 4.8h. Given such a significant gap in computational cost, a direct performance comparison with other methods in the main table would not be entirely fair. We have included this discussion in Section 5.4.
>
> **Requested change 4.**
>
> We thank the reviewer for raising this point. Our claim is grounded in the design of ReciNet. In the local module, graph edge features depend only on interatomic distances. In the reciprocal module, the representation is built from fractional coordinates and reciprocal lattice vectors. Together, they provide a representation that is consistent under equivalent lattice descriptions.
>
> To empirically verify this beyond architectural reasoning, we provide an analysis in which we apply equivalent transformations to crystals and measure the resulting prediction differences. As shown in the table below, the result aligns with the theoretical design of the model.
>
> Table: Prediction consistency under equivalent crystal representations on JARVIS Bandgap (MBJ). We report the absolute difference between the prediction on the original crystal and on equivalent transformations.
>
> | Equivalent representation | Absolute prediction difference (eV) |
> |---|---:|
> | Rigid rotation ($120^\circ$ about the [111] axis) | 0.0010 |
> | Translation shift | 0.0008 |
>
> **Requested change 5.**
>
> We thank the reviewer for this suggestion. We acknowledge that parts of our original discussion were phrased a bit strongly, and we agree that the appendix analysis could be presented as a suggestive interpretability analysis rather than a definitive physical validation. Accordingly, we have revised this section to soften the claims and more carefully qualify the conclusions drawn from the learned reciprocal-space filters.

---

### Review · Reviewer_A3Xe · 2026-04-14

**Summary Of Contributions:**

This paper proposed ReciNet, a neural network + featurization scheme which predicts properties of periodic crystals by combining a GNN over their spatial arrangement of atoms with features from their Fourier transform, aligned to the unit cell instead of in Cartesian space as in previous work.

**Audience:**

Yes

**Audience Explanation:**

Yes, there are lots of people in ML working on materials and crystal structures.

**Broader Impact Concerns:**

No concerns.

**Claims And Evidence:**

Yes

**Claims Explanation:**

The answer is _mostly_ yes: the method is described reasonably clearly and the experiments make sense. However, Tables 1-2 are effectively making a claim of statistical significance without any measure of variation. Often the differences between the top few methods is small in absolute terms, within what I suspect is the magnitude of random variation between reruns of the same neural network training method. In such cases, I don't believe strong claims about performance differences are supported. I think the authors need to either support the ranking claims with some kind of measurement of variation, or remove the claims of "best method", "second best method", etc.

**Requested Changes:**

- Add statistical significance or remove claims of relative performance in Table 1-2
- Can you comment on the invariance/equivariance properties of Recinet? One thing that I noticed was that you use Euclidean distances in the real-space GNN, but unit cell coordinates in reciprocal space. Am I correct that this means it won't be translationally invariant, or at least not invariant to a change of basis of the unit cell? Given the strong interest in equivariant/invariant methods in this community, I think at least some sort of statement about these properties is warranted.
- Fourier transform methods effectively assume infinite periodicity, and of course real materials are not infinite. By using reciprocal space features ReciNet seems especially prone to model misspecification due to this effect. Can you comment on this? Is this likely to affect results in practice? It might be worth mentioning in the limitations section.

---

> ### Author Response · Authors · 2026-04-17
>
> **Requested change 1.**
>
> We appreciate this suggestion. In the revised manuscript, we include deviation results for ReciNet on both the JARVIS and Materials Project datasets in Appendix B.10 (Tables 17 and 18). The results suggest that ReciNet training is stable and that the reported results are not sensitive to random initialization.
> We have revised the manuscript to more precisely characterize our empirical results in the abstract, introduction, and our experimental summary (Section 5.1). We believe this revised manuscript provides a more balanced presentation with respect to Tables 1–3.
>
> **Requested change 2.**
>
> ReciNet is designed to preserve invariance under equivalent crystal representations. In the local module, graph construction and edge features depend only on interatomic Euclidean distances, and are therefore invariant to rigid translations and rotations. In the reciprocal module, the representation is built from fractional coordinates together with reciprocal lattice vectors.
>
> More specifically, the reciprocal interaction is expressed through Fourier phase terms. Under a global translation of the crystal, the forward and inverse Fourier phases acquire matching offsets that cancel, so the resulting update remains unchanged. Likewise, under a consistent change of basis of the unit cell, the fractional coordinates and reciprocal basis transform jointly, leaving the phase term unchanged.
>
> **Requested change 3.**
>
> We thank the reviewer for this question. We address it from three perspectives.
> First, the infinite periodicity assumption is standard in this domain, not specific to ReciNet. The crystal representation $M=(A,F,L)$ adopted by baselines in this domain (CGCNN, ALIGNN, Matformer, PotNet, ComFormer, Crystalformer) defines crystals via unit cells that generate infinite periodic structures. Crucially, the DFT calculations that provide the ground-truth labels for these benchmarks are performed under periodic boundary conditions. ReciNet is therefore not more prone to misspecification relative to competing methods.
> Second, ReciNet's hybrid design provides inherent robustness. The local geometric GNN module captures short-range bonding independently of periodicity assumptions, serving as a complementary pathway. Third, our ionic–covalent analysis (Table 6) provides direct evidence: on covalent materials where long-range periodicity is less relevant ($\Delta X < 0.5$), ReciNet remains fully competitive (within 1.3\% of baselines), showing the reciprocal module does not introduce noticeable degradation.
> Lastly, the infinite periodicity assumption becomes less appropriate in real-world materials, such as nanoscale crystallites, highly defective crystals, or systems with broken periodicity. We have added a statement in the limitations section.

---

> > ### Comment · Reviewer_A3Xe · 2026-04-23
> > **Overall happy with changes, but please make some small revisions**
> >
> > Thank you for the updates and sorry for the delayed reply.
> >
> > **Requested change 1 (statistical significance)**: two comments:
> >
> > - Tables 17/18 are not at all referenced in the main text. Could you include a clearer reference? For example, in the caption for Tables 1-2, you could include the phrase "measures of variation in Table N".
> > - Tables 17/18 are only one half of the story- you report the deviation for ReciNet, but none of the baselines. It is not enough for ReciNet's performance to be stable- for ReciNet to be _better_ than another method its performance would also need to be stable. I'm not explicitly asking for you to test all the baselines, but what about the following:
> >   - Weakening the claims of "best performance"
> >   - Re-running the benchmark for _one_ of the baselines (e.g. iComFormer or CrystalFramer)
> >
> > **Requested change 2 (statistical significance)**: that makes sense, thanks for answering.
> >
> > **Requested change 3 (infinite periodicity)**: thanks for adding the extra sentence. It's a minor point but good to add a few words.

---

> > > ### Author Response · Authors · 2026-04-27
> > >
> > > **Requested change 1.**
> > >
> > > We thank the reviewer for this suggestion. We have added explicit references in Section 5.1 at the end of the Materials Project and JARVIS-DFT result paragraphs to refer to the variation results. To further address the question, we reran eComFormer on all JARVIS-DFT tasks, as reported below. The results show that ReciNet maintains strong performance even with variation considered.
> > >
> > > **Mean $\pm$ standard deviation for eComFormer on all tasks in the JARVIS dataset.**
> > >
> > > |  | Form. Energy | Bandgap(OPT) | $E_{\text{total}}$ | Bandgap(MBJ) | $E_{\text{hull}}$ |
> > > |---|---|---|---|---|---|
> > > |  | meV/atom | eV | meV/atom | eV | meV |
> > > | ReciNet | 27.0$\pm$0.04 | 0.126$\pm$0.0002 | 26.8$\pm$0.04 | 0.24$\pm$0.01 | 43$\pm$0.03 |
> > > | eComFormer | 28.7$\pm$0.06 | 0.124$\pm$0.0002 | 32.3$\pm$0.05 | 0.28$\pm$0.01 | 44$\pm$0.05 |
> > >
> > > **Requested changes 2&3.**
> > >
> > > We thank the reviewer for the feedback and are glad that our clarifications address the requested changes 2 and 3.

---

### Decision · Action_Editor_jawB · 2026-05-10

**Recommendation:** Accept as is

**Additional Comments:**

The paper introduces ReciNet. The architecture combines enriches a GNN with reciprocal blocks that use learnable filters to perform Fourier transform of the lattice representation (based on fractional coordinates and reciprocal lattice representation). The modification is natural and delivers strong performance across commonly used benchmarks.

Discussion has largely focused on the experimental section and the statistical significance of the claims. Authors have done a good job addressing the feedback.

All reviewers voted for accepting the paper. The reviewers appreciated the solid physical motivation and the strong empirical coverage across standard benchmarks (JARVIS, Materials Project, and MatBench).

All in all, it is my pleasure to recommend acceptance of the work.

**Audience:**

Yes

**Audience Explanation:**

The paper will likely be of interest to researchers interested in modeling material properties. Using long-ranged interactions is a natural approach that has the potential to see uptake in the subfield.

**Claims And Evidence:**

Yes

**Claims Explanation:**

The key claims regarding performance are well supported. Reviewers suggested softening certain claims and adding uncertainty estimates, which has been addressed during the discussion period. There is sufficient evidence, especially after the discussion period, that the architecture is better at modeling long-range interactions.